# Adversarial Counterfactual Learning and Evaluation for Recommender System

**Da Xu, Chuanwei Ruan** *
Walmart Labs, Sunnyvale, CA 94086
{Da.Xu, Chuanwei.Ruan}@walmartlabs.com

**Evren Korpeoglu, Sushant Kumar, Kannan Achan**
Walmart Labs, Sunnyvale, CA 94086
{EKorpeoglu, SKumar4, KAchan}@walmartlabs.com

## Abstract

The feedback data of recommender systems are often subject to what was exposed to the users; however, most learning and evaluation methods do not account for the underlying exposure mechanism. We first show in theory that applying supervised learning to detect user preferences may end up with inconsistent results in the absence of exposure information. The counterfactual propensity-weighting approach from causal inference can account for the exposure mechanism; nevertheless, the partial-observation nature of the feedback data can cause identifiability issues. We propose a principled solution by introducing a minimax empirical risk formulation. We show that the relaxation of the dual problem can be converted to an adversarial game between two recommendation models, where the opponent of the candidate model characterizes the underlying exposure mechanism. We provide learning bounds and conduct extensive simulation studies to illustrate and justify the proposed approach over a broad range of recommendation settings, which shed insights on the various benefits of the proposed approach.

## 1 Introduction

In the offline learning and evaluation of recommender systems, the dependency of feedback data on the underlying exposure mechanism is often overlooked. When the users express their preferences on the products explicitly (such as providing ratings) or implicitly (such as clicking), the feedback are conditioned on the products to which they are exposed. In most cases, the previous exposures are decided by some underlying mechanism such as the history recommender system. The dependency causes two dilemmas for machine learning in recommender systems, and solutions have yet been found satisfactorily. Firstly, the majority of supervised learning models only handle the dependency between label (user feedback) and features, yet in the actual feedback data, the exposure mechanism can alter the dependency pathways (Figure 1). In Section 2, we show from a theoretical perspective that directly applying supervised learning on feedback data can result in inconsistent detection of the user preferences. Secondly, an unbiased model evaluation should have the product exposure determined by the candidate recommendation model, which is almost never satisfied using the feedback data only. The second dilemma also reveals a major gap between evaluating models by online experiments and using history data, since the offline evaluations are more likely to bias toward the history exposure mechanism as it decided to what products the users might express their preferences. The disagreement between the online and offline evaluations may partly explain the

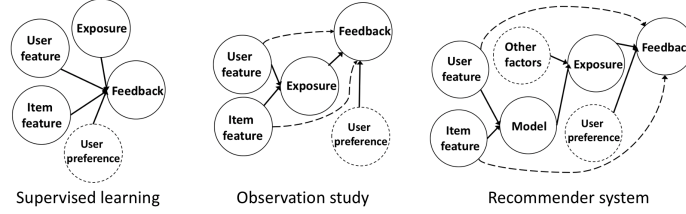

Figure 1: The graphical model representation for the causal inference view under different settings.

controversial observations made in several recent papers, where deep recommendation models are overwhelmed by classical collaborative filtering approaches in offline evaluations [7, 28], despite their many successful deployments in the real-world applications [5, 6, 40, 44, 43, 42].

To settle the above dilemmas for recommender systems, we refer to the idea of *counterfactual* modelling from the observational studies and causal inference literature [22, 24, 30] to redesign the learning and evaluation methods. Briefly put, the counterfactual modelling answers questions related to "what if", e.g. what is the feedback data if the candidate model were deployed. Our key purpose of introducing the counterfactual methods is to take account of the dependency between the feedback data and exposure. Relevant proposals have been made in several recent papers [32, 20, 17, 1, 21, 39, 13]; however, most of them rely on excessive data or model assumptions (such as the missing-data model we describe in Section 2) that may not be satisfied in practice. Many of the assumptions are essentially unavoidable due to a fundamental discrepancy between the recommender system and observational studies. In observational studies, the exposure (treatment) status are fully observed, and the exposure mechanism is completely decided by the covariates (features) [29, 3]. For recommender systems, the exposure is only partially captured by the feedback data. The complete exposure status can only be retrieved from the system's backend log, whose access is highly restricted, and rarely exists for the public datasets. Also, the exposure mechanism can depend on intractable randomness, e.g. burst events, special offers, interference with other modules such as the advertisement, as well as the relevant features that are not attainable from feedback data. In Figure 1, we show the causal diagrams for the three different views of recommender system. A direct consequence of the above differences is that the exposure mechanism is not *identifiable* from feedback data, i.e. we can modify the conditional distribution characterized by the exposure mechanism without disturbing the observation distribution. Therefore, the existing methods have to make problem-specific or unjustifiable assumptions in order to bypass or simply ignore the identifiability issue.

Our solution is to acknowledge the uncertainty brought by the identifiability issue and treat it as an adversarial component. We propose a minimax setting where the candidate model is optimized over the worst-case exposure mechanism. By applying duality arguments and relaxations, we show that the minimax problem can be converted to an *adversarial game* between two recommendation models. Our approach is entirely novel and principled. We conclude the contributions as follow.

- We provide the first theoretical analysis to show an inconsistent issue of supervised learning on recommender systems, caused by the unknown exposure mechanism.

- We propose a minimax setting for counterfactual recommendation and convert it to a tractable two-model adversarial game. We prove the generalization bounds for the proposed adversarial learning, and provide analysis for the minimax optimization.

- We carry out extensive simulation and real data experiments to demonstrate our performance, and deploy online experiments to fully illustrate the benefits of the proposed approach.

## 2   Preliminaries

We use bold-faced letters to denote vectors and matrices, upper-case letters to denote random variables and the corresponding lower-case letters to denote observations. Distributions are denoted by $P$ and $Q$. Let $\mathbf{x}_u$ be the user feature vector for user $u \in \{1, \ldots, n\}$, $\mathbf{z}_i$ be the item feature vector for item $i \in \{1, \ldots, m\}$, $O_{u,i} \in \{0, 1\}$ be the exposure status, $Y_{u,i}$ be the feedback and $\mathcal{D}$ be the collected user-item pairs where non-positive interactions may come from negative sampling. The feature vectors can be one-hot encoding or embedding, so our approach is fully compatible with

deep learning models that leverage representation learning and are trained under negative sampling. Recommendation models are denoted by such as $f_\theta$ and $g_\psi$. They take $\mathbf{x}_u$, $\mathbf{z}_i$ (and the exposure $o_{u,i}$ if available) as input. We use the shorthand $f_\theta(u, i)$ to denote the output score, and the loss with respect to the $y_{u,i}$ is given by $\delta(y_{u,i}, f_\theta(u, i))$. Our notations also apply to the sequential recommendation by encoding the previously-interacted items to the user feature vector $\mathbf{x}$.

We use $p_g(O_{u,i}|\mathbf{x}_u, \mathbf{z}_i)$ to denote the exposure mechanism that depends on the underlying model $g$. Also, $p(Y_{u,i}|O_{u,i}, \mathbf{x}_u, \mathbf{z}_i)$ gives the user response, which is independent from the exposure mechanism whenever $O_{u,i}$ is observed. We point out that the stochasticity in the exposure can also be induced by the exogenous factors (unobserved confounders) who bring extra random perturbations. We do not explicitly differentiate the explcit and implicit feedback setting unless specified.

**Supervised learning for feedback data.**

Let $Y_{u,i} \in \{-1, 1\}$ be the implicit feedback. Set aside the exposure for a moment, the goal of supervised learning is to determine the optimal recommendation function that minimizes the surrogate loss: $\ell_\phi(f_\theta) = \frac{1}{|\mathcal{D}|} \sum_{(u,i) \in \mathcal{D}} [\phi(Y_{u,i} \cdot f_\theta(u, i))]$, where $\phi$ induces the widely-adopted margin-based loss. Now we take account of the (unobserved) exposure status by first letting:

$$p^{(1)}(o) = p(Y_{u,i} = 1, O_{u,i} = o, \mathbf{x}_u, \mathbf{z}_i), \ p^{(-1)}(o) = p(Y_{u,i} = -1, O_{u,i} = o, \mathbf{x}_u, \mathbf{z}_i), \ o \in \{0, 1\},$$

to denote the joint distribution for positive and negative feedback under either exposure status. The surrogate loss, which now depends on $p^{(1)}$ and $p^{(-1)}$ since we include the exposure, is denoted by $L_\phi(f_\theta, \{p^{(1)}, p^{(-1)}\})$. In the following claim, we show that if we fix the exposure mechanism and optimize $f_\theta$, the optimal loss and the corresponding $f_\theta^*$ depend only on $p^{(1)}$ and $p^{(-1)}$.

**Claim 1.** *When the exposure mechanism $p(O_{u,i}|\mathbf{X}_u, \mathbf{Z}_i)$ is **given and fixed**, the optimial loss is:*

$$\inf_{f_\theta} L_\phi(f_\theta, \{p^{(1)}, p^{(-1)}\}) = -D_c(P^{(1)}||P^{(-1)}), \tag{1}$$

*where $P^{(1)}$ and $P^{(-1)}$ are the corresponding distributions for $p^{(1)}$ and $p^{(-1)}$, and $D_c(P^{(1)}||P^{(-1)}) = \int c(\frac{p^{(1)}}{p^{(-1)}}) dP^{(-1)}$ is the f-divergence induced by the convex, lower-semicontinuous function $c$. Also, the optimal $f_\theta^*$ that achieves the infimum is given by $\alpha_\phi^*(\frac{p^{(1)}}{p^{(-1)}})$ for some function $\alpha_\phi^*$ that depends on $\phi$.*

We defer the proof to the appendix. Notice that the joint distribution can be factorized into: $p(Y_{u,i}, o_{u,i}, \mathbf{x}_u, \mathbf{x}_i) \propto p(Y_{u,i}|o_{u,i}, \mathbf{x}_u, \mathbf{z}_i) \cdot p_g(o_{u,i}|\mathbf{x}_u, \mathbf{z}_i)$, so Claim 1 implies that:

$$f_\theta^*(\mathbf{x}_u, \mathbf{z}_i; o_{u,i}) = \alpha_\phi^*\Big(p(Y_{u,i} = 1|o_{u,i}, \mathbf{x}_u, \mathbf{z}_i) \big/ p(Y_{u,i} = -1|o_{u,i}, \mathbf{x}_u, \mathbf{z}_i)\Big).$$

We conclude that: **1.** when the exposure mechanism is given, the optimal loss $-D_c(P^{(1)}||P^{(-1)})$ is a function of both the user preference and the exposure mechanism; **2.** the optimal model $f_\theta^*$ depends only on the user preference, since $f_\theta^*$ is a function of $p(Y|o, \mathbf{x}, \mathbf{z})$ which does not depend on the exposure mechanism (mentioned at the beginning of this section). Both conclusions are practically reasonable, as the optimal recommendation model should only detect user preference regardless of the exposure mechanisms. The optimal loss, on the other hand, depends on the joint distribution where the underlying exposure mechanism plays a part.

However, when $p(O_{u,i}|\mathbf{X}_u, \mathbf{Z}_i)$ is unknown, the conclusions from Claim 1 no longer hold and the optimal $f_\theta^*$ will depend on the exposure mechanism. As a consequence, if the same feedback data were collected under different exposure mechanisms, the recommendation model may find the user preference differently. The inconsistency is caused by not accounting for the unknown exposure mechanism from the supervised learing. We mention that another line of research studies the user preference and exposure in an interactive online fashion using such as the contextual bandit and reinforcement learning [19, 43]. The discussions of which are beyond the scope of this paper.

**The propensity-weighting approach**.

In causal inference, the probability of exposure given the observed features (covariates) is referred to as the propensity score [29]. The propensity-weighting approach uses weights based on the propensity score to create a synthetic sample in which the distribution of observed features is independent of exposure [15, 3]. It especially appeals to us because we want the feedback data to

be made independent of the exposure mechanism. The propensity-weighted loss is constructed via: $\frac{1}{|\mathcal{D}|} \sum_{(u,i) \in \mathcal{D}} \phi\big(y_{u,i} \cdot f_\theta(\mathbf{x}_u, \mathbf{z}_i)\big) \big/ p(O_{u,i} = 1|\mathbf{x}_u, \mathbf{z}_i)$, and by taking the expectation with respect to exposure (whose distribution is denoted by $Q$), we recover the ordinary loss:

$$\mathbb{E}_Q\Big[\frac{1}{|\mathcal{D}|} \sum_{(u,i) \in \mathcal{D}} \frac{\phi\big(y_{u,i} \cdot f_\theta(\mathbf{x}_u, \mathbf{z}_i)\big)}{p(O_{u,i} = 1|\mathbf{x}_u, \mathbf{z}_i)}\Big] = \mathbb{E}_{P_n}\Big[\frac{\phi\big(Y \cdot f_\theta(\mathbf{X}, \mathbf{Z})\big)}{p(O = 1|\mathbf{X}, \mathbf{Z})} p(O = 1|\mathbf{X}, \mathbf{Z})\Big] = \ell_\phi(f_\theta), \quad (2)$$

where the second expectation is taken with respect to the empirical distribution $P_n$. Let $Q_0$ be the distribution for the underlying exposure mechanism. The propensity-weighted empirical distribution is then given by $P_n/Q_0$ (after scaling), which can be think of as the synthetic sample distribution after eliminating the influence from the underlying exposure mechanism. It is straightforward to verify that after scaling, the expected propensity-weighted loss is exactly given by: $\mathbb{E}_{P_n/Q_0}\big[\phi(Y \cdot f_\theta(\mathbf{X}, \mathbf{Z}))\big]$.

**The hidden assumption of the missing-data (click) model**

A number of prior work deals with the unidentifiable exposure mechanism by assuming a missing-data model [31, 2, 21, 37], which is also referred to as the *click model*:

$$p(\text{click} = 1|x) = p(\text{expose} = 1|x) \cdot p(\text{relevance} = 1|x). \quad (3)$$

While the *click model* greatly simplifies the problem since the exposure mechanism can now be characterized explicitly, it relies on a hidden assumption that is rarely satisfied in practice. We use $R$ to denote the relevance and $Y$ to denote the click. The fact that $Y = 1 \Leftrightarrow O = 1$ and $R = 1$ implies:

$$p(Y = 1|x) = p(O = 1, R = 1|x) = p(O = 1|x) \cdot p(R = 1|O = 1, x)$$

$$\overset{(3)}{\Longrightarrow} p(R = 1|O = 1, x) = p(R = 1|x),$$

which suggests that being relevant is independent of getting exposed given the features. This is rarely true (or at least cannot be examined) in many real-world problems, unless $x$ contains every single factor that may affect the exposure and user preference. We aim at providing a robust solution whenever the hidden assumption of the missing-data (click) model is dubious or violated.

## 3  Method

Let $P^*$ be the ideal exposure-eliminated sample distribution corresponding to $P/Q_0$, according to the underlying exposure mechanism $Q_0$ and data distribution $P$. For notation simplicity, without overloading the original meaning by too much, from this point we treat $P$, $P_n$, $Q_0$ and $P^*$ as distributions on the sample space $\mathcal{X}$ which consists of all the observed data $(\mathbf{x}_u, \mathbf{z}_i, y_{u,i})$ with $(u, i) \in \mathcal{D}$. Since we make no data or model assumptions that may allow us to accurately recover $P^*$, we introduce a minimax formulation to characterize the uncertainty. We optimize $f_\theta$ against the worst possible choice of (a hypothetical) $\hat{P}$, whose discrepancy with the ideal $P^*$ can only be determined by the data to a neighborhood: $\text{Dist}(P^*, \hat{P}) < \rho$. Among the divergence and distribution distance measures, we choose the Wasserstein distance for our problem, which is defined as:

$$W_c(\hat{P}, P^*) = \inf_{\gamma \in \Pi(\hat{P}, P^*)} \mathbb{E}_{((\mathbf{x},\mathbf{z},y),(\mathbf{x}',\mathbf{z}',y')) \sim \gamma}\big[c\big((\mathbf{x}, \mathbf{z}, y), (\mathbf{x}', \mathbf{z}', y')\big)\big], \quad (4)$$

where $c : \mathcal{X} \times \mathcal{X} \to [0, +\infty)$ is the convex, lower semicontinuous transportation cost function with $c(\mathbf{t}, \mathbf{t}) = 0$, and $\Pi(\hat{P}, P^*)$ is the set of all distributions whose marginals are given by $\hat{P}$ and $P^*$. Intuitively, the Wasserstein distance can be interpreted as the minimum cost associated with transporting mass between probability measures. We choose the Wasserstein distance instead of others exactly because we wish to understand how to transport from the empirical data distribution to an ideal synthetic data distribution where the observations were independent of the exposure mechanism. Hence, we consider the local minimax *empirical risk minimization (ERM)* problem:

$$\underset{f_\theta \in \mathcal{F}}{\text{minimize}} \ \underset{W_c(P^*, \hat{P}) < \rho}{\sup} \ \mathbb{E}_{\hat{P}}\big[\delta(Y, f_\theta(\mathbf{X}, \mathbf{Z}))\big], \quad (5)$$

where we directly account for the uncertainty induced by the lack of identifiability in the exposure mechanism, and optimize $f_\theta$ under the worst possible setting. However, the formulation in (5) is first of all a constraint optimization problem. Secondly, the constraint is expressed in terms of the hypothetical $P^*$. After applying a duality argument, we express the dual problem via the exposure mechanism in the following Claim 2. We use $\hat{Q}$ to denote some estimation of $Q_0$.

**Claim 2.** *Suppose that the transportation cost $c$ is continuous and the propensity score are all bounded away from zero, i.e. $p(O_{i,u} = 1|\mathbf{x}_u, \mathbf{z}_i) \geq \mu$. Let $\mathcal{P} = \{P : W_c(P^*, P) < \rho\}$, then*

$$\sup_{\hat{P} \in \mathcal{P}} \mathbb{E}_{\hat{P}}\big[\delta(Y, f_\theta(\mathbf{X}, \mathbf{Z}))\big] = \inf_{\alpha \geq 0} \Big\{\alpha\rho + \sup_{\hat{Q}} \Big\{\mathbb{E}_P \Big[\frac{\delta(Y, f_\theta(\mathbf{X}, \mathbf{Z}))}{\hat{q}(O = 1|\mathbf{X}, \mathbf{Z})}\Big] - c_0\alpha W_c(\hat{Q}^{-1}, Q_0^{-1})\Big\}\Big\},$$

*where $c_0$ is a positive constant and $\hat{q}$ is the density function associated with $\hat{Q}$.*

We defer the proof to the appendix. If we consider the relaxation for each fixed $\alpha$ (see the appendix), the minimax objective has a desirable formulation where $\alpha$ becomes a tuning parameter:

$$\operatorname*{minimize}_{f_\theta \in \mathcal{F}} \sup_{\hat{Q}} \mathbb{E}_P\Big[\frac{\delta(Y, f_\theta(\mathbf{X}, \mathbf{Z}))}{\hat{q}(O = 1|\mathbf{X}, \mathbf{Z})}\Big] - \alpha W_c(\hat{Q}, Q_0), \quad \alpha \geq 0. \tag{6}$$

To make sense of (6), we see that while $\hat{Q}$ is acting adversarially against $f_\theta$ as the inverse weights in the first term, it cannot arbitrarily increase the objective function, since the second terms acts as a regularizer that keeps $\hat{Q}$ close to the true exposure mechanism $Q_0$. Compared with the primal problem in (5), the relaxed dual formulation in (6) gives the desired unconstrained optimization problem. Also, we point out that the exposure mechanism is often given by the recommender system that was operating during the data collection, which we shall leverage as a domain knowledge to further convert (6) to a more tractable formulation. Let $g^*$ be the recommendation model that underlies $Q_0$. Assume for now that $p_g(O = 1|\mathbf{X}, \mathbf{Q})$ is given by $G\big(g(\mathbf{X}, \mathbf{Z})\big) \in (\mu, 1), \mu > 0$ for some transformation function $G$. We leave the inclusion and manipulation of the unobserved factors to Section 3.2. The objective in (6) can then be converted to a two-model adversarial game:

$$\operatorname*{minimize}_{f_\theta \in \mathcal{F}} \sup_{g_\psi \in \mathcal{G}} \mathbb{E}_P\Big[\frac{\delta(Y, f_\theta(\mathbf{X}, \mathbf{Z}))}{G\big(g_\psi(\mathbf{X}, \mathbf{Z})\big)}\Big] - \alpha W_c(G(g_\psi), G(g^*)), \quad \alpha \geq 0. \tag{7}$$

Before we move on to discuss the implications of (7), its practical implementations and the minimax optimization, we first show and discuss the theoretical guarantees for the generalization error, in comparison to the standard ERM setting, after introducing the adversarial component.

## 3.1 Theoretical property

Before we state the main results, we need to characterize the loss function corresponding to the adversarial objective as well as the complexity of our hypothesis space. For the first purpose, we introduce the cost-regulated loss which is defined as: $\Delta_\gamma\big(f_\theta; (\mathbf{x}, \mathbf{z}, y)\big) = \sup_{(\mathbf{x}', \mathbf{z}', y') \in \mathcal{X}} \Big\{\frac{\delta\big(y', f_\theta(\mathbf{x}', \mathbf{z}')\big)}{q(o=1|\mathbf{x}', \mathbf{z}')} - \gamma c\big((\mathbf{x}, \mathbf{z}, y), (\mathbf{x}', \mathbf{z}', y')\big)\Big\}$, For the second purpose, we consider the *entropy integral* $\mathcal{J}(\tilde{\mathcal{F}}) = \int_0^\infty \sqrt{\log \mathcal{N}(\epsilon; \tilde{\mathcal{F}}, \|.\|_\infty)} d\epsilon$, where $\tilde{\mathcal{F}} = \{\delta(f_{\theta,\cdot})|f_\theta \in \mathcal{F}\}$ is the hypothesis class and $\mathcal{N}(\epsilon; \tilde{\mathcal{F}}, \|\cdot\|_\infty)$ gives the *covering number* for the $\epsilon$−cover of $\tilde{\mathcal{F}}$ in terms of the $\|\cdot\|_\infty$ norm. Suppose that $|\delta(y, f_\theta(\mathbf{x}, \mathbf{z}))| \leq M$ holds uniformly. Now we state our main theoretical result on the worst-case generalization bound under the minimax setting, and the proof is delegated to the appendix.

**Theorem 1.** *Suppose the mapping $G$ from $g_\psi$ to $q(o = 1|\mathbf{x}, \mathbf{z})$ is one-to-one and surjective with $g_\psi \in \mathcal{G}$. Let $\tilde{\mathcal{G}}(\rho) = \big\{g_\psi \in \mathcal{G} \,|\, W_c\big(G(g_\psi), G(g^*)\big) \leq \rho\big\}$. Then under the conditions specified in Claim 2, for all $\gamma \geq 0$ and $\rho > 0$, the following inequality holds with probability at least $1 - \epsilon$:*

$$\sup_{g_\psi \in \tilde{\mathcal{G}}(\rho)} \mathbb{E}_P\Big[\frac{\delta(Y, f_\theta(\mathbf{X}, \mathbf{Z}))}{G\big(g_\psi(\mathbf{X}, \mathbf{Z})\big)}\Big] \leq c_1\gamma\rho + \mathbb{E}_{P_n}[\Delta_\gamma\big(f_\theta; (\mathbf{X}, \mathbf{Z}, Y)\big)] + \frac{24\mathcal{J}(\tilde{\mathcal{F}}) + c_2(M, \sqrt{\log \frac{2}{\epsilon}}, \gamma)}{\sqrt{n}},$$

*where $c_1$ is a positive constants and $c_2$ is a simple linear function with positive weights.*

The above generalization bound holds for all $\rho$ and $\delta$, and we show that when they are decided by some data-dependent quantities, the result can be converted to some simplified forms that reveal the more direct connections with the propensity-weighted loss and standard ERM results.

**Corollary 1.** *Following the statements in Theorem 1, there exists some data-dependent $\gamma_n$ and $\rho_n(f_\theta)$, such that when $\gamma \geq \gamma_n$, for all $\rho > 0$:*

$$Pr\Big( \sup_{g_\psi \in \tilde{\mathcal{G}}(\rho)} \mathbb{E}_P\Big[\frac{\delta(Y, f_\theta(\mathbf{X}, \mathbf{Z}))}{G\big(g_\psi(\mathbf{X}, \mathbf{Z})\big)}\Big] \leq c_1\gamma\rho + \mathbb{E}_{P_n}\Big[\frac{\delta\big(f_\theta; (\mathbf{X}, \mathbf{Z}, Y)\big)}{q(O = 1|\mathbf{X}, \mathbf{Z})}\Big] + \varepsilon_n(\epsilon) \Big) > 1 - \epsilon;$$

*and when $\rho = \rho_n(f_\theta)$, for all $\gamma \geq 0$:*

$$Pr\Big( \sup_{g_\psi \in \tilde{\mathcal{G}}(\rho)} \mathbb{E}_P\Big[\frac{\delta(Y, f_\theta(\mathbf{X}, \mathbf{Z}))}{G\big(g_\psi(\mathbf{X}, \mathbf{Z})\big)}\Big] \leq \sup_{P:W_c(P,P_n)\leq\tilde{\rho}} \mathbb{E}_P\Big[\frac{\delta\big(f_\theta; (\mathbf{X}, \mathbf{Z}, Y)\big)}{q(O = 1|\mathbf{X}, \mathbf{Z})}\Big] + \varepsilon_n(\epsilon) \Big) > 1 - \epsilon,$$

*where $\varepsilon_n(\epsilon) = \big(24\mathcal{J}(\tilde{\mathcal{F}}) + c_2(M, \sqrt{\log\frac{2}{\epsilon}}, \gamma)\big)/\sqrt{n}$ as suggested by Theorem 1.*

Corollary 1 shows that the proposed approach has the same $1/\sqrt{n}$ rate as the standard ERM. Also, the first result reveals an extra $\delta\rho$ bias term induced by the adversarial setting, the second result characterizes how the additional uncertainty is reflected on the propensity-weighted empirical loss.

### 3.2 Practical implementations

Directly optimizing the minimax objective in (7) is infeasible since $g^*$ is unknown and the Wasserstein distance is hard to compute when $\mathcal{G}$ is a complicated model such as neural network [23]. Nevertheless, understanding the comparative roles of $f_\theta$ and $g_\psi$ can help us construct practical solutions.

Recall that our goal is to optimize $f_\theta$. The auxiliary $g_\psi$ is introduced to characterize the adversarial exposure mechanism, so we are less interested in recovering the true $g^*$. With that being said, the term $W_c(G(g_\psi), G(g^*))$ only serves to establish certain regularizations on $g_\psi$ such that it is constrained by the underlying exposure mechanism. Relaxing or tightening the regularization term should not significantly impact the solution since we can always adjust the regularization parameter $\alpha$. Hence, we are motivated to design tractable regularizers to approximate or even replace $W_c(G(g_\psi), G(g^*))$, as long as the constraint on $g_\psi$ is established under the same principle. Similar ideas have also been applied to train the *generative adversarial network (GAN)*: the optimal classifier depends on the unknown data distribution, so in practice, people use alternative tractable classifiers that fit into the problem [11]. We list several alternative regularizers for $g_\psi$ as below.

- In the explicit feedback data setting, the exposure status is partially observed, so the loss of $G(g_\psi)$ on the partially-observed exposure data can be used as the regularizer, i.e. $\frac{1}{|\mathcal{D}_{\exp}|} \sum_{(u,i)\in\mathcal{D}_{\exp}} \phi\big(g_\psi(\mathbf{x}_u, \mathbf{z}_i)\big)$, where $\mathcal{D}_{\exp} = \{(u,i) \in \mathcal{D}|o_{u,i} = 1\}$.
- For the content-based recommendations, the exposure often have high correlation with popularity where the popular items are more likely to be recommended. So the regularizer may leverage the empirical popularity via: $\text{corr}\big(\frac{1}{m}\sum_u G(g_\psi(\mathbf{X}_u, \mathbf{Z}_i)), \frac{1}{m}\sum_u Y_{u,i}\big)$.
- In the implicit feedback setting, if all the other choices are impractical, we may simply use the loss on the feedback data as a regularizer: $\mathbb{E}_{P_n}\big[\phi(Y \cdot g_\psi(\mathbf{X}, \mathbf{Z}))\big]$. The loss-based regularizer is meaningful because $g^*$ is often determined by some other recommendation models. If it happens that $g^* \in \mathcal{G}$, we can expect similar performances from $g_\psi$ and $g^*$ on the same feedback data since the exposure mechanism is determined by $g^*$ itself.

We focus on the third example because it applies to almost all cases without requiring excessive assumptions. Therefore, the practical adversarial objective is now given by:

$$\underset{f_\theta \in \mathcal{F}}{\text{minimize}} \sup_{g_\psi \in \mathcal{G}} \mathbb{E}_{P_n}\Big[\frac{\delta(Y, f_\theta(\mathbf{X}, \mathbf{Z}))}{G\big(g_\psi(\mathbf{X}, \mathbf{Z})\big)}\Big] - \alpha\mathbb{E}_{P_n}\big[\delta(Y, g_\psi(\mathbf{X}, \mathbf{Z}))\big], \quad \alpha \geq 0. \tag{8}$$

In the next step, we study how to handle the unobserved factors who also play a part in the exposure mechanism. As we mentioned in Section 1, having unobserved factors is inevitable practically. In particular, we leverage the *Tukey's factorization* proposed in the missing data literature [9]. In the presence of unobserved factors, Tukey's factorization suggests that we additionally characterize the relationship between exposure mechanism and outcome [8] (see the appendix for detailed discussions). Relating the outcome to exposure mechanism has also been found in the recommendation literature [32]. For clarity, we employ a simple logistic-regression to model $G$ as:

$$G_\beta\big(g_\psi(\mathbf{x}, \mathbf{z}), y\big) = \sigma\big(\beta_0 + \beta_1 g_\psi(\mathbf{x}, \mathbf{z}) + \beta_2 y\big),$$

where $\sigma(\cdot)$ is the sigmoid function. We now reach the final form of the adversarial game:

$$\underset{f_\theta \in \mathcal{F}, \beta}{\text{minimize}} \sup_{g_\psi \in \mathcal{G}} \mathbb{E}_{P_n}\left[\frac{\delta(Y, f_\theta(\mathbf{X}, \mathbf{Z}))}{G_\beta(g_\psi(\mathbf{X}, \mathbf{Z}), Y)}\right] - \alpha \mathbb{E}_{P_n}\left[\delta(Y, g_\psi(\mathbf{X}, \mathbf{Z}))\right], \quad \alpha \geq 0. \tag{9}$$

We place $\beta$ to the minimization problem for the following reason. By our design, $G_\beta$ merely characterizes the potential impact of unobserved factors which we do not consider to act adversarially. Otherwise, the adversarial model can be too strong for $f_\theta$ to learn anything useful.

### 3.3 Minimax optimization and robust evaluation

---
**Algorithm 1:** Minimax optimization
---
**Input:** Learning rates $r_\theta, r_\psi$,
      discounts $d_\theta, d_\psi > 1$;
**while** *loss not stabilized* **do**
    $\theta = \theta - r_\theta \mathbb{E}_{\text{batch}} \nabla_\theta \ell(f_\theta, g_\psi)$;
    $\psi = \psi + r_\psi \mathbb{E}_{\text{batch}} \nabla_\psi \ell(f_\theta, g_\psi)$;
    $r_\theta = r_\theta/d_\theta, r_\psi = r_\psi/d_\psi$;
**end**

---

To handle the adversarial training, we adopt the sequential optimization setup where the players take turn to update their model. Without loss of generality, we treat the objective in (8) as a function of of the two models: $\min_{f_\theta} \max_{g_\psi} \ell(f_\theta, g_\psi)$. When $\ell$ is nonconvex-nonconcave, the classical Minimax Theorem no longer hold and $\min_{f_\theta} \max_{g_\psi} \ell(f_\theta, g_\psi) \neq \max_{g_\psi} \min_{f_\theta} \ell(f_\theta, g_\psi)$ [34]. Consequently, which player goes first has important implications. Here, we choose to train $f_\theta$ first because $g_\psi$ can then choose the worst candidate from the uncertainty set in order to undermines $f_\theta$.

We adopt the two-timescale gradient descent ascent (GDA) [14] schema that is widely applied to train adversarial objectives (Algorithm 1). However, the existing analysis on GDA's converging to local Nash equilibrium assumes simultaneous training [14, 27, 25], so their guarantees do not apply here. Instead, we keep training until the objective stops changing by updating either $f_\theta$ or $g_\psi$.

Consequently, the stationary points in Algorithm 1 may not attain local Nash equilibrium. Nevertheless, when the timescale of the two models differ significantly (by adjusting the initial learning rates and discounts), it has been shown that the stationary points belong to the *local minimax solution* up to some degenerate cases [16]. The local minimaxity captures the *optimal strategies* in the sequential game if both models are only allowed to change their strategies locally. Hence, Algorithm 1 leads to solutions that are locally optimal. Finally, the role of $G_\beta$ is less important in the sequential game, and we do not observe significant differences from updating it before or after $f_\theta$ and $g_\psi$.

Recommenders are often evaluated by the *mean square error (MSE)* on explicit feedback, and by the information retrieval metric such as DCG and NDCG on implicit feedback. After the training, we obtain the candidate model $f_\theta$ as well as the $G_\beta(g_\psi)$ who gives the worst-case propensity score function specialized for $f_\theta$. Therefore, instead of pursuing unbiased evaluation, we instead consider the *robust evaluation* by using $G_\beta(g_\psi)$. It frees the offline evaluation from the potential impact of exposure mechanism, and thus provide a robust view on the true performance. For instance, the robust NDCG can be computed via: $\frac{1}{|\mathcal{D}_{\text{test}}|} \sum_{(u,i) \in \mathcal{D}_{\text{test}}} \text{NDCG}(y_{u,i}, f_\theta(\mathbf{x}_u, \mathbf{z}_i))/G_\beta(g_\psi(\mathbf{x}_u, \mathbf{z}_i))$.

## 4 Relation to other work

The propensity-weighting method is proposed and intensively studied in the observation studies and causal inference literature [3, 4]. A recent work that introduces adversarial training to solve the identifiability issue studies on the covariate-balancing methods [18, 41]. Adversarial training is widely applied by such as generative models [11], model defense [35], adversarial robustness [38] and distributional robust optimization (DRO) [26]. Compare with GAN, we study the sampling distribution instead of the generating distribution, and GAN does not involve counterfactual modelling. DRO often focus on the feature distribution while we study the propensity score distribution. Using the Wasserstein distance as regularization is also common in the literature [33, 10]. Here, we introduce the adversarial setting for the identifiability issue, whereas the model defense and adversarial robustness study the training and modelling properties under deliberate adversarial behaviors.

Counterfactual modelling for recommenders often relies on certain data or model assumptions (such as the click model assumption) to make up for the identifiability issue, and is thus venerable when the assumptions are violated in practice [32, 20, 17, 1, 21, 39, 13, 31, 2, 37]. Adversarial training for recommenders often borrows the GAN setting by assuming a generative distribution for certain components [36, 12]. Here, we do not assume the generative nature of recommender systems.

|  | MLP | MLP | MLP | GMF | GMF | GMF | **ACL-MLP** | **ACL-GMF** |
| **config** | Pop | MLP | Oracle | Pop | GMF | Oracle | MLP | GMF |
| *MovieLens-1M* | | | | | | | | |
| Hit@10 | 39.60 (.12) | 39.24 (.3) | 39.68 (.3) | 39.00 (.2) | 39.47 (.1) | 39.10 (.2) | **40.32** (.1) | 39.08 (.2) |
| NDCG@10 | 20.26 (.1) | 20.10 (.2) | 20.33 (.2) | 19.33 (.3) | 19.58 (.2) | 19.30 (.1) | **20.81** (.2) | 19.61 (.1) |
| *Goodreads* | | | | | | | | |
| Hit@10 | 31.90 (.3) | 30.61 (.2) | **33.82** (.1) | 30.01 (.3) | 31.36 (.2) | 33.50 (.1) | 33.45 (.2) | 32.51 (.2) |
| NDCG@10 | 16.65 (.2) | 15.72 (.2) | **17.81** (.1) | 15.24 (.2) | 16.40 (.2) | 17.28 (.2) | 17.50 (.1) | 16.85 (.1) |

| Data | *MovieLens-1M* | | | | *Goodreads* | | | |
| Model | Pop | CF | MLP | GMF | Pop | CF | MLP | GMF |
| Hit@10 | 33.76 (.1) | 38.27 (.2) | 39.43 (.2) | 39.00 (.2) | 26.62 (.3) | 30.90 (.2) | 31.78 (.2) | 29.59 (.3) |
| NDCG@10 | 17.75 (.1) | 18.59 (.2) | 20.09 (.2) | 19.28 (.3) | 14.29 (.2) | 16.43 (.1) | 16.58 (.2) | 14.94 (.2) |

Table 1: Unbiased evaluations (using the true exposure) for the baselines and the proposed approach on the semi-synthetic data. **Upper panel:** we provide in the **config** rows the $g_\psi$ model (such as using the baseline models and the oracle model) when trained with the propensity-score (PS) approach or the proposed approach (marked by the **ACL-**). **Lower panel:** the original baseline models without using propensity-score approach or ACL. We use bold-font and underscore to mark the best and second-best outcomes. The mean and standard deviation are computed over ten repetitions, and the complete numerical results are deferred to the appendix.

# 5    Experiment and Result

We conduct simulation study, real-data analysis, as well as online experiments to demonstrate the various benefits of the proposed adversarial counterfactual learning and evaluation approach.

- In the simulation study, we generate the synthetic data using real-world explicit feedback dataset so that we have access to the oracle exposure mechanism. We then show that models trained by our approach achieve superior unbiased offline evaluation performances.

- In the real-world data analysis, we demonstrate that the models trained by our approach also achieve more improvements even using the standard offline evaluation.

- By conducting online experiments, we verify that our robust evaluation is more accurate than the standard offline evaluation when compared with the actual online evaluations.

As for the baseline models, since we are proposing a high-level learning and evaluation approach that are compatible with almost all the existing recommendation models, we consider the well-known baseline models to demonstrate the effectiveness of our approach. Specifically, we employ the popularity-based recommendation (**Pop**), matrix factorization collaborative filtering (**CF**), multi-layer perceptron-based CF model (**MLP**), neural CF (**NCF**) and the generalized matrix factorization (**GMF**), as the representatives for the content-based recommendation. We also consider the prevailing attention-based model (**Attn**) as a representative for the sequential recommendation. We also choose $f_\theta$ and $g_\psi$ among the above baselines models for our adversarial counterfactual learning. To fully demonstrate the effectiveness of the proposed adversarial training, we also experiment with the non-adversarially trained propensity-score method **PS**, where we first optimize $g_\psi$ only on the regularization term until convergence, keep it fixed, and then train $f_\theta$ in the regular propensity-weighted ERM setting. For the sake of notation, we refer to our learning approach as the **ACL-**.

We choose to examine the various methods with the widely-adopted next-item recommendation task. In particular, all but the last two user-item interactions are used for training, the second-to-last interaction is used for validation, and the last interaction is used for testing. All the detailed data processing, experiment setup, model configuration, parameter tuning, training procedure, validation, testing and sensitivity analysis are provided in the appendix.

**Synthetic data analysis**. We use the explicit feedback data from *MovieLens-1M*[2] and *Goodreads* datasets. We train a baseline CF model and use the optimized hidden factors to generate a synthetic exposure mechanism (with the details presented in the appendix), and treat it as the *oracle* exposure. The implicit feedback data are then generated according to the oracle exposure as well as the optimized hidden factors. Unbiased offline evaluation is now possible because we have access to the

|  | Pop | CF | MLP | NCF | GMF | Attn | PS | **ACL** | **ACL** |
|---|---|---|---|---|---|---|---|---|---|
| | | | | ***MovieLens-1M*** | | | | | |
| **config** | | | | | | | Attn / Pop | GMF / GMF | Attn / Attn |
| Hit@10 | 42.18 (.2) | 60.97 (.1) | 61.01 (.2) | 63.37 (.3) | 63.97(.1) | 82.66 (.2) | 81.97 (.1) | 64.32 (.2) | **83.64** (.1) |
| NDCG@10 | 21.99 (.1) | 32.59 (.1) | 32.09 (.3) | 33.49 (.1) | 33.82(.2) | 55.27 (.1) | 54.51 (.1) | 33.70 (.1) | **55.71** (.2) |
| | | | | ***LastFM*** | | | | | |
| **config** | | | | | | | GMF / Pop | GMF / GMF | Attn /Attn |
| Hit@10 | 25.26 (.2) | 52.97 (.3) | 81.86 (.3) | 81.87 (.3) | 83.12 (.3) | 71.89 (.3) | 82.64 (.2) | **83.64** (.2) | 72.02 (.2) |
| NDCG@10 | 15.35 (.1) | 31.54 (.2) | 58.38 (.2) | 57.33 (.4) | 58.96 (.2) | 59.75 (.2) | 58.84 (.2) | 59.11 (.1) | **59.45** (.1) |
| | | | | ***Goodreads*** | | | | | |
| **config** | | | | | | | Attn / Pop | GMF / GMF | Attn / Attn |
| Hit@10 | 43.36 (.1) | 60.32 (.2) | 62.17 (.2) | 63.11 (.3) | 63.78 (.1) | 72.63 (.2) | 73.39 (.1) | 64.17 (.2) | **73.82** (.3) |
| NDCG@10 | 22.73 (.2) | 37.73 (.1) | 37.65 (.1) | 38.78 (.3) | 38.69 (.1) | 48.98 (.1 ) | 49.92 (.3) | 39.53 (.1) | **49.99** (.1) |

Table 2: Standard evaluations (without accounting for exposure) for the baselines and proposed approach on the benchmark data. Similarly, we provide in the **config** rows the $f_\theta$ and $g_\psi$ model choice when trained with the PS and our ACL approach. We present here the best $f_\theta$ and $g_\psi$ combination for the PS method, and the full results for our approach and the baselines are deferred to the appendix.

exposure mechanism. Also, to set a reasonable benchmark under our simulation setting, we provide the additional experiments where $g_\psi$ is given by the oracle exposure model. The results are provided in Table 1. We see that when trained with the proposed approach, the baselines models yield their best performances (other than the oracle-enhanced counterparts) under the unbiased offline evaluation, and outperforms the rest of the baselines, which reveals the first appeal of our approach.

**Real data analysis.** Other than using the *MovieLens-1M* and *Goodreads* data in the implicit feedback setting, we further include the *LastFM* music recommendation (implicit feedback) dataset. From the results in Table 2, we observe that the models trained by our approach achieve the best outcome, even using the standard evaluation where the exposure mechanism is not considered. The better performance in standard evaluation suggests the second appeal of the adversarial counterfactual learning, that even though it optimizes towards the minimax setting, the robustness is not at the cost of the performance under the standard evaluation.

| MSE on metric | Standard | Popularity debiased | Propensity model debiased | Robust |
|---|---|---|---|---|
| Hit@5 | .18(.10) | .14(.08) | .14(.06) | **.12**(.04) |
| NDCG@5 | .10(.06) | .09(.05) | .08(.05) | **.07**(.03) |

Table 3: The mean-squared error (MSE) to online evaluation results from eight online experiments.

**Online experiment analysis.** To examine the practical benefits of the proposed robust learning and evaluation approach in real-world experiments, we carry out several online A/B testings on the *Walmart.com*, a major e-commerce platform in the U.S., in a content-based item recommendation setting. We are provided with the actual online testing and evaluation results. All the candidate models were trained offline using the proposed approach. We compare the standard offline evaluation, popularity-debiased offline evaluation (where the item popularity is used as the propensity score), the propensity-score model approach and our robust evaluation, with respect to the actual online evaluations. In Table 3, we see that our proposed evaluation approach is indeed a more robust approximation to the online evaluation. It reveals the third appeal of the proposed approaches that they are capable of narrowing the gap between online and offline evaluations.

## 6   Conclusion

We thoroughly analyze the drawback of supervised learning for recommender systems and propose the theoretically-grounded adversarial counterfactual learning and evaluation framework. We provide elaborated theoretical and empirical results to illustrate the benefits of the proposed approach.
**Scope and limitation**. The improvement brought by our approach ultimately depends on the properties of the feedback data, e.g. to what extent is the identifiability issue causing uncertainties in the data. Also, we observe empirically that the propensity model can experience undesired behaviors during the adversarial training as a consequence of using suboptimal tuning parameters. Therefore, it remains to be studied how the optimization dynamics can impact the two-model interactions for the proposed adversarial counterfactual learning.

## Broader Impact

To the best of our knowledge, the approaches discussed in this paper raise no major ethical concerns and societal consequences. Researchers and practitioners from the recommender system domain may benefit from our research since robust offline learning and evaluation has been a significant challenge in real-world applications. The worst possible outcome when the proposed approach fails is that it reduces to the standard offline learning as the propensity model stops making the desired impact. Finally, the proposed approach aims at solving the identifiability issues of the data, the extent of which depends on the properties of the data.

## Acknowledgments and Disclosure of Funding

The work is supported by the Walmart U.S. eCommerce. The authors declare that there is no conflict of interest.

## Footnotes

*Both authors contribute equally to this work.

[2]All the data sources, processing steps and other detailed descriptions are provided in the appendix.

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
