[Supplementary Material]

# Appendix

We provide in the Appendix proofs for the major theoretical results. We also discuss the relaxation for Claim 2, and show the implications from the Tukey's factorization on unobserved factors that leads to our final objective in (8). We present the experiment details and the complete numerical results, including demonstrations on the adversarial training process of the two models.

## A.1 Proof for Claim 1

*Proof.* When taking the exposure mechanism into account, minimizing $f_\theta$ over the loss is implicitly doing $\inf_{f_\theta} L_\phi(f_\theta, \{p^{(1)}, p^{(-1)}\})$, where

$$
\begin{aligned}
L_\phi(f_\theta, \{p^{(1)}, p^{(-1)}\}) &= \mathbb{E}\Big[\phi\big(Y \cdot f_\theta(\mathbf{x}, \mathbf{z}; O)\big)\Big] \\
&= \sum_{o \in \{0,1\}} \phi\big(f_\theta(\mathbf{x}, \mathbf{z}; O = o)\big)p^{(1)}(o) + \phi\big(-f_\theta(\mathbf{x}, \mathbf{z}; O = o)\big)p^{(-1)}(o).
\end{aligned}
$$

For any fixed exposure mechanism $p(O|\mathbf{x}, \mathbf{z})$, we have

$$
\begin{aligned}
\inf_{f_\theta} L_\phi(f_\theta, \{p^{(1)}, p^{(-1)}\}) &= \sum_{o \in \{0,1\}} \inf_\alpha \big\{\phi(\alpha)p^{(1)}(o) + \phi(-\alpha)p^{(-1)}(o)\big\} \\
&= \sum_{o \in \{0,1\}} p^{(1)}(o) \inf_\alpha \Big\{\phi(\alpha) + \phi(-\alpha)\frac{p^{(-1)}(o)}{p^{(1)}(o)}\Big\}.
\end{aligned} \tag{A.1}
$$

For each $o \in \{0, 1\}$, let $\mu(o) = p^{(-1)}(o)/p^{(1)}(o)$ and $\Delta(\mu) = -\inf_\alpha \big(\phi(\alpha) + \phi(-\alpha\mu)\big)$. Notice that $\Delta(\mu)$ is a convex function of $\mu$ since the supremum (negative of the infimum) over a set of affine functions is convex. Since $\Delta$ is convex and continuous, we get:

$$
\inf_{f_\theta} L_\phi(f_\theta, \{p^{(1)}, p^{(-1)}\}) = -\sum_{o \in \{0,1\}} p^{(1)}(o)\Delta\Big(\frac{p^{(-1)}(o)}{p^{(1)}(o)}\Big),
$$

which is exactly the f-divergence $D_\Delta(P^{(1)}||P^{(-1)})$ induced by $\Delta$. Also, up on achieving the infimum in (A.1), the optimal $f_\theta$ is given by solving $a_\phi^*(\mu) = \arg\min_\alpha \big(\phi(\alpha) + \phi(-\alpha)\mu\big)$. $\qquad\square$

## A.2 Proof for Claim 2 and the relaxation

We first proved the result in Claim 2 and then show the corresponding relaxation given by (5).

*Proof.* For the estimation $\hat{P} = P/\hat{Q}$ of the ideal exposure-eliminated sample, $W_c(\hat{P}, P^*) \leq \rho$ is equivalent to $W_c\big(P/\hat{Q}, P/Q_0\big) \leq \rho$. The key observation is that when $P$ is given by the empirical distribution that assigns uniform weights to all samples, the Wasserstein's distance $W_c\big(P/\hat{Q}, P/Q_0\big)$ is convex in $\hat{Q}^{-1}$ (since $c$ is convex) and $\hat{Q} = Q_0$ gives $W_c\big(P/\hat{Q}, P/Q_0\big) = 0$. Since we assume that the propensity scores are all bounded away from zero, so $P/\hat{Q}$ and $P/Q_0$ exist and and have

normal behavior. So we able to establish the duality results, since the Slater's condition holds. Let $\mathbf{h} = (\mathbf{x}, \mathbf{z}, y) \in \mathcal{X}$ and $\mathcal{X}'$ be a copy of $\mathcal{X}$. We have

$$\sup_{\hat{P}:W_c(\hat{P},P^*)\leq\rho} \int \delta\big(y, f_\theta(\mathbf{x}, \mathbf{z})\big) d\hat{P}(\mathbf{h})$$

$$= \sup_{\hat{Q}:W_c\big(P/\hat{Q},P/Q_0\big)\leq\rho} \int \frac{\delta\big(y, f_\theta(\mathbf{x}, \mathbf{z})\big)}{\hat{q}(O = 1 \,|\, \mathbf{x}, \mathbf{z})} d\hat{Q}(\mathbf{h})$$

$$= \inf_{\alpha\geq 0} \sup_{\hat{Q}} \left\{ \int \frac{\delta\big(y, f_\theta(\mathbf{x}, \mathbf{z})\big)}{\hat{q}(O = 1 \,|\, \mathbf{x}, \mathbf{z})} d\hat{Q}(\mathbf{h}) - \alpha W_c\big(P/\hat{Q}, P/Q_0\big) + \alpha\rho \right\} \tag{A.2}$$

$$= \inf_{\alpha\geq 0} \sup_{\hat{Q}} \left\{ \int \frac{\delta\big(y, f_\theta(\mathbf{x}, \mathbf{z})\big)}{\hat{q}(O = 1 \,|\, \mathbf{x}, \mathbf{z})} d\hat{Q}(\mathbf{h}) - \alpha \inf_{\gamma\in\Pi\big(P/\hat{Q},P/Q_0\big)} \int c(\mathbf{h}, \mathbf{h}') d\gamma(\mathbf{h}, \mathbf{h}') + \alpha\rho \right\}$$

$$= \inf_{\alpha\geq 0} \sup_{\hat{Q}} \sup_{\gamma\in\Pi\big(P/\hat{Q},P/Q_0\big)} \left\{ \int \Big(\frac{\delta_{f_\theta}(\mathbf{h})}{\hat{q}(\mathbf{h})} - \alpha c(\mathbf{h}, \mathbf{h}')\Big) d\gamma(\mathbf{h}, \mathbf{h}') + \alpha\rho \right\},$$

where in the last line we use the shorthand notation $\delta_{f_\theta}(\mathbf{h}) := \delta\big(y, f_\theta(\mathbf{x}, \mathbf{z})\big)$ and $\hat{q}(\mathbf{h}) := \hat{q}(O = 1|\mathbf{x}, \mathbf{z})$. Then notice that

$$\sup_{\hat{Q}} \sup_{\gamma\in\Pi\big(\frac{P}{\hat{Q}},\frac{P}{Q_0}\big)} \int \Big(\frac{\delta_{f_\theta}(\mathbf{h})}{\hat{q}(\mathbf{h})} - \alpha c(\mathbf{h}, \mathbf{h}')\Big) d\gamma(\mathbf{h}, \mathbf{h}') \leq \int \sup_{\mathbf{h}\in\mathcal{X}} \Big(\frac{\delta_{f_\theta}(\mathbf{h})}{\hat{q}(\mathbf{h})} - \alpha c(\mathbf{h}, \mathbf{h}')\Big) dQ_0(\mathbf{h}'),$$

$$\tag{A.3}$$

and we then show that the opposite direction also holds so it is always equality. Let $\mathcal{K}$ be the space of measurable conditional distributions (Markov kernels) from $\mathcal{X}$ to $\mathcal{X}'$, then

$$\sup_{\hat{Q}} \sup_{\gamma\in\Pi\big(\frac{P}{\hat{Q}},\frac{P}{Q_0}\big)} \int \Big(\frac{\delta_{f_\theta}(\mathbf{h})}{\hat{q}(\mathbf{h})} - \alpha c(\mathbf{h}, \mathbf{h}')\Big) d\gamma(\mathbf{h}, \mathbf{h}')$$

$$\geq \sup_{K\in\mathcal{K}} \int \Big(\frac{\delta_{f_\theta}(\mathbf{h})}{\hat{q}(\mathbf{h})} - \alpha c(\mathbf{h}, \mathbf{h}')\Big) dK(\mathbf{h} \,|\, \mathbf{h}') dQ_0(\mathbf{h}'). \tag{A.4}$$

In the next step, we consider the space of all measurable mappings $\mathbf{h}' \mapsto \mathbf{h}(\mathbf{h}')$ from $\mathcal{X}'$ to $\mathcal{X}$, denoted by $\mathcal{H}$. Since all the mappings are measurable, the underlying spaces are regular, and $\delta_{f_\theta}$ and $c$ are at least semi-continuous, using standard measure theory arguments for exchanging the integration and supremum, we get

$$\sup_{\mathbf{h}(\cdot)\in\mathcal{H}} \int \Big(\frac{\delta_{f_\theta}\big(\mathbf{h}(\mathbf{h}')\big)}{\hat{q}\big(\mathbf{h}(\mathbf{h}')\big)} - \alpha c\big(\mathbf{h}(\mathbf{h}'), \mathbf{h}'\big)\Big) dQ_0(\mathbf{h}') = \int \sup_{\mathbf{h}\in\mathcal{X}} \Big(\frac{\delta_{f_\theta}(\mathbf{h})}{\hat{q}(\mathbf{h})} - \alpha c(\mathbf{h}, \mathbf{h}')\Big) dQ_0(\mathbf{h}'),$$

$$\tag{A.5}$$

where the $\mathbf{h}(\cdot)$ on the LHS represents the mapping, and the $\mathbf{h}$ on the RHS still denotes elements from the sample space $\mathcal{X}$. Now we let the support of the conditional distribution $K(\mathbf{h} \,|\, \mathbf{h}')$ given by $\mathbf{h}(\mathbf{h}')$. So according to (A.5), we have:

$$\sup_{K\in\mathcal{K}} \int \Big(\frac{\delta_{f_\theta}(\mathbf{h})}{\hat{q}(\mathbf{h})} - \alpha c(\mathbf{h}, \mathbf{h}')\Big) dK(\mathbf{h} \,|\, \mathbf{h}') dQ_0(\mathbf{h}')$$

$$= \sup_{\mathbf{h}(\cdot)\in\mathcal{H}} \int \Big(\frac{\delta_{f_\theta}\big(\mathbf{h}(\mathbf{h}')\big)}{\hat{q}\big(\mathbf{h}(\mathbf{h}')\big)} - \alpha c\big(\mathbf{h}(\mathbf{h}'), \mathbf{h}'\big)\Big) dQ_0(\mathbf{h}')$$

$$\geq \int \sup_{\mathbf{h}\in\mathcal{X}} \Big(\frac{\delta_{f_\theta}(\mathbf{h})}{\hat{q}(\mathbf{h})} - \alpha c(\mathbf{h}, \mathbf{h}')\Big) dQ_0(\mathbf{h}') \tag{A.6}$$

$$\geq \sup_{\hat{Q}} \sup_{\gamma\in\Pi\big(\frac{P}{\hat{Q}},\frac{P}{Q_0}\big)} \int \Big(\frac{\delta_{f_\theta}(\mathbf{h})}{\hat{q}(\mathbf{h})} - \alpha c(\mathbf{h}, \mathbf{h}')\Big) d\gamma(\mathbf{h}, \mathbf{h}').$$

Combining (A.6), (A.4) and (A.3), we see that

$$\sup_{\hat{Q}} \sup_{\gamma \in \Pi\left(\frac{P}{\hat{Q}}, \frac{P}{Q_0}\right)} \int \left(\frac{\delta_{f_\theta}(\mathbf{h})}{\hat{q}(\mathbf{h})} - \alpha c(\mathbf{h}, \mathbf{h}')\right) d\gamma(\mathbf{h}, \mathbf{h}') = \int \sup_{\mathbf{h} \in \mathcal{X}} \left(\frac{\delta_{f_\theta}(\mathbf{h})}{\hat{q}(\mathbf{h})} - \alpha c(\mathbf{h}, \mathbf{h}')\right) dQ_0(\mathbf{h}').$$

(A.7)

Finally, notice that

$$\sup_{\hat{Q}} \sup_{\gamma \in \Pi\left(\frac{P}{\hat{Q}}, \frac{P}{Q_0}\right)} \int \left(\frac{\delta_{f_\theta}(\mathbf{h})}{\hat{q}(\mathbf{h})} - \alpha c(\mathbf{h}, \mathbf{h}')\right) d\gamma(\mathbf{h}, \mathbf{h}') = \sup_{\hat{Q}} \int \frac{\delta_{f_\theta}(\mathbf{h})}{\hat{q}(\mathbf{h})} d\hat{Q}(h) - \alpha W_c\left(P/\hat{Q}, P/Q_0\right),$$

so according to (A.2), we reach the final result:

$$\sup_{\hat{P}: W_c(\hat{P}, P^*) \leq \rho} \int \delta\big(y, f_\theta(\mathbf{x}, \mathbf{z})\big) d\hat{P}(\mathbf{h}) = \inf_{\alpha \geq 0} \left\{ \alpha\rho + \int \sup_{\mathbf{h} \in \mathcal{X}} \left(\frac{\delta_{f_\theta}(\mathbf{h})}{\hat{q}(\mathbf{h})} - \alpha c(\mathbf{h}, \mathbf{h}')\right) dQ_0(\mathbf{h}') \right\}$$

$$= \inf_{\alpha \geq 0} \left\{ \alpha\rho + \sup_{\hat{Q}} \int \frac{\delta_{f_\theta}(\mathbf{h})}{\hat{q}(\mathbf{h})} d\hat{Q}(h) - \alpha W_c\big(P/\hat{Q}, P/Q_0\big) \right\}.$$

(A.8)

$\square$

To reach the relaxation given in (5), we use the alternate expression for the Wasserstein distance obtained from the Kantorovich-Rubinstein duality [8]. We denote the Lipschitz continuity for a function $f$ by $\|f\|_{L \leq l}$. When the cost function $c$ is $l$-Lipschitz continuous, $W_c(P_1, P_2)$ is also referred to as the Wasserstein-$l$ distance. Without loss of generality, we consider $\|c\|_{L \leq 1}$ such as the $\ell_2$ norm, and with that the Wasserstein distance is equivalent to:

$$W_c\big(P/\hat{Q}, P/Q_0\big) = \sup_{\|f\|_{L \leq 1}} \big\{ \mathbb{E}_{\mathbf{h} \sim P/\hat{Q}} f(\mathbf{h}) - \mathbb{E}_{\mathbf{h} \sim P/Q_0} f(\mathbf{h}) \big\},$$

(A.9)

where $f : \mathcal{X} \to \mathbb{R}$. In practice, when $P$ is the empirical distribution that assigns uniform weights to all the samples, we have

$$W_c\big(P_n/\hat{Q}, P_n/Q_0\big) = \sup_{\|f\|_{L \leq 1}} \big\{ \mathbb{E}_{\mathbf{h} \sim P_n/\hat{Q}} f(\mathbf{h}) - \mathbb{E}_{\mathbf{h} \sim P_n/Q_0} f(\mathbf{h}) \big\}$$

$$= \sup_{\|f\|_{L \leq 1}} \left\{ a_1 \mathbb{E}_{\mathbf{h} \sim P_n} \frac{f(\mathbf{h})}{\hat{q}(\mathbf{h})} - a_2 \mathbb{E}_{\mathbf{h} \sim P_n} \frac{f(\mathbf{h})}{q_0(\mathbf{h})} \right\}$$

$$= \sup_{\|f\|_{L \leq 1}} \mathbb{E}_{\mathbf{h} \sim P_n} \left[ \frac{f(\mathbf{h})}{\hat{q}(\mathbf{h}) \cdot q_0(\mathbf{h})} \big(a_1 q_0(\mathbf{h}) - a_2 \hat{q}(\mathbf{h})\big) \right]$$

$$\leq \sup_{\mathbf{h} \in \mathcal{X}} \left\{ \frac{1}{\hat{q}(\mathbf{h}) \cdot q_0(\mathbf{h})} \right\} \cdot \sup_{\|f\|_{L \leq 1}} \big\{ a_3 \mathbb{E}_{\mathbf{h} \sim P_n \cdot Q_0} f(\mathbf{h}) - a_4 \mathbb{E}_{\mathbf{h} \sim P_n \cdot \hat{Q}} f(\mathbf{h}) \big\}$$

$$\leq \frac{1}{\mu^2} \sup_{\|f\|_{L \leq \max\{a_5, a_6\}}} \big\{ \mathbb{E}_{\mathbf{h} \sim Q_0} f(\mathbf{h}) - \mathbb{E}_{\mathbf{h} \sim \hat{Q}} f(\mathbf{h}) \big\}$$

$$= \frac{1}{\mu^2} W_{\tilde{c}}(\hat{Q}, Q_0),$$

(A.10)

where the-above $a_i$ are all constants induced by using the change-of-measure with important-weighting estimators, and the induced cost function $\tilde{c}$ on the last line satisfies $\|\tilde{c}\|_{L \leq \max\{a_5, a_6\}}$. Therefore, we see that the Wasserstein distance between $P_n/\hat{Q}$ and $P_n/Q_0$ can be bounded by $W_{\tilde{c}}(\hat{Q}, Q_0)$. Hence, for each $\alpha \geq 0$ in (A.8),

$$\sup_{\hat{Q}} \mathbb{E}_P \left[ \frac{\delta(Y, f_\theta(\mathbf{X}, \mathbf{Z}))}{\hat{q}(O = 1 | \mathbf{X}, \mathbf{Z})} \right] - \tilde{\alpha} W_{\tilde{c}}(\hat{Q}, Q_0), \quad \tilde{\alpha} \geq 0,$$

is a relaxation of the result in Claim 2. In practice, the specific forms of the cost functions $c$ or $\tilde{c}$ do not matter, because the Wasserstein distance is intractable and we use the data-dependent surrogates that we discuss in Section 3.2.

## A.3 Proof for Theorem 1

*Proof.* Following the same arguments from the proof in Claim 2, we obtain the similar result stated in (A.8) that

$$
\begin{aligned}
&\sup_{g_\psi \in \tilde{\mathcal{G}}(\rho)} \mathbb{E}_P\left[\frac{\delta(Y, f_\theta(\mathbf{X}, \mathbf{Z}))}{G(g_\psi(\mathbf{X}, \mathbf{Z}))}\right] \\
&\leq \inf_{\gamma \geq 0}\left\{\gamma\rho + \int \sup_{\mathbf{h}\in\mathcal{X}}\left(\frac{\delta_{f_\theta}(\mathbf{h})}{\hat{q}(\mathbf{h})} - \gamma c(\mathbf{h}, \mathbf{h}')\right)dP(\mathbf{h})\right\} \\
&= \inf_{\gamma \geq 0}\left\{\gamma\rho + \mathbb{E}_P\left[\Delta_\gamma(f_\theta; \mathbf{H})\right]\right\} \quad \text{(by the definition of } \Delta_\gamma) \\
&\leq \inf_{\gamma \geq 0}\left\{\gamma\rho + \mathbb{E}_{P_n}\left[\Delta_\gamma(f_\theta; \mathbf{H})\right] + \sup_{f_\theta \in \mathcal{F}}\left(\mathbb{E}_P\left[\Delta_\gamma(f_\theta; \mathbf{H})\right] - \mathbb{E}_{P_n}\left[\Delta_\gamma(f_\theta; \mathbf{H})\right]\right)\right\}.
\end{aligned}
\tag{A.11}
$$

Let $W_\gamma = \sup_{f_\theta \in \mathcal{F}}\left(\mathbb{E}_P\left[\Delta_\gamma(f_\theta; \mathbf{H})\right] - \mathbb{E}_{P_n}\left[\Delta_\gamma(f_\theta; \mathbf{H})\right]\right)$, then notice that

$$
W_\gamma = \frac{1}{n}\sup_{f_\theta \in \mathcal{F}}\left[\sum_{i=1}^N \mathbb{E}_P\left[\Delta_\gamma(f_\theta; \mathbf{H})\right] - \Delta_\gamma(f_\theta; \mathbf{H}_i)\right] \quad \gamma \geq 0.
$$

Since $|\delta_{f_\theta}(\mathbf{h})| \leq \mu M$ holds uniformly, according to the McDiarmid's inequality on bounded random variables, we first have

$$
p\left(W_\gamma - \mathbb{E}W_\gamma \geq \mu M\sqrt{\frac{\log 1/\epsilon}{2N}}\right) \leq \epsilon.
\tag{A.12}
$$

Then let $\epsilon_1, \ldots, \epsilon_N$ be the i.i.d Rademacher random variables independent of $\mathbf{H}$, and $\mathbf{H}_i'$ be the i.i.d copy of $\mathbf{H}_i$ for $i = 1, \ldots, N$. Applying the symmetrization argument, we see that

$$
\begin{aligned}
\mathbb{E}W_\gamma &= \mathbb{E}\left[\sup_{f_\theta \in \mathcal{F}}\left|\sum_{i=1}^N \Delta_\gamma(f_\theta; \mathbf{H}_i') - \sum_{i=1}^N \Delta_\gamma(f_\theta; \mathbf{H}_i)\right|\right] \\
&= \mathbb{E}\left[\sup_{f_\theta \in \mathcal{F}}\left|\frac{1}{N}\sum_{i=1}^N \epsilon_i\Delta_\gamma(f_\theta; \mathbf{H}_i') - \frac{1}{N}\sum_{i=1}^N \Delta_\gamma(f_\theta; \mathbf{H}_i)\right|\right] \\
&\leq 2\mathbb{E}\left[\sup_{f_\theta \in \mathcal{F}}\left|\frac{1}{N}\sum_{i=1}^N \epsilon_i\Delta_\gamma(f_\theta; \mathbf{H}_i)\right|\right].
\end{aligned}
\tag{A.13}
$$

It is clear that each $\epsilon_i\Delta_\gamma(f_\theta; \mathbf{H}_i)$ is zero-mean, and now we show that it is sub-Gaussian as well. For any two $f_\theta, f_\theta'$, we show the bounded difference:

$$
\begin{aligned}
&\mathbb{E}\left[\exp\left(\lambda\left(\frac{1}{\sqrt{N}}\epsilon_i\Delta_\gamma(f_\theta; \mathbf{H}_i) - \frac{1}{\sqrt{N}}\epsilon_i\Delta_\gamma(f_\theta'; \mathbf{H}_i)\right)\right)\right] \\
&= \left(\mathbb{E}\left[\exp\left(\frac{\lambda}{\sqrt{N}}\epsilon_1\left(\Delta_\gamma(f_\theta; \mathbf{H}_1) - \Delta_\gamma(f_\theta'; \mathbf{H}_1)\right)\right)\right]\right)^N \\
&= \left(\mathbb{E}\left[\exp\left(\frac{\lambda}{\sqrt{N}}\epsilon_1\left(\sup_{\mathbf{h}'}\inf_{\mathbf{h}''}\left\{\frac{\delta_{f_\theta}(\mathbf{h}')}{q(\mathbf{h}')} - \gamma c(\mathbf{H}_1, \mathbf{h}') - \frac{\delta_{f_\theta'}(\mathbf{h}'')}{q(\mathbf{h}'')}\right\} + \gamma c(\mathbf{H}_1, \mathbf{h}'')\right)\right)\right]\right)^N \\
&\leq \left(\mathbb{E}\left[\exp\left(\frac{\lambda}{\sqrt{N}}\epsilon_1\left(\sup_{\mathbf{h}'}\left\{\frac{\delta_{f_\theta}(\mathbf{h}')}{q(\mathbf{h}')} - \frac{\delta_{f_\theta'}(\mathbf{h}')}{q(\mathbf{h}')}\right\}\right)\right)\right]\right)^N \\
&\leq \exp\left(\lambda^2\left\|\frac{\delta_{f_\theta}}{q} - \frac{\delta_{f_\theta'}}{q}\right\|_\infty^2/2\right) \quad \text{(by Hoeffding's inequality).}
\end{aligned}
\tag{A.14}
$$

Hence we see that $\frac{1}{\sqrt{N}}\epsilon_i\Delta_\gamma(f_\theta; \mathbf{H}_i)$ is sub-Gaussian with respect to $\left\|\frac{\delta_{f_\theta}}{q} - \frac{\delta_{f_\theta'}}{q}\right\|_\infty^2$. Therefore, $\mathbb{E}W_\gamma$ can be bounded by using the standard technique for Rademacher complexity and Dudley's entropy integral [7]:

$$
\mathbb{E}W_\gamma \leq \frac{24}{N}\mathcal{J}(\tilde{\mathcal{F}}).
\tag{A.15}
$$

Combining all the above bounds in (A.11), (A.12) and (A.15) we obtain the desired result. $\qquad\square$

## A.4 Proof for Corollary 1

*Proof.* To obtain the first result, let the data-dependent $\gamma_n$ be given by

$$\gamma_n = \max_i \sup_{\mathbf{h}' \in \mathcal{H}} \left( \frac{\delta_{f_\theta}(\mathbf{h}')}{q(\mathbf{h}')} - \frac{\delta_{f_\theta}(\mathbf{h}_i)}{q(\mathbf{h}_i)} \right) \Big/ c(\mathbf{h}_i, \mathbf{h}').$$

Then according to the definition of $\Delta_\gamma$, we have

$$\mathbb{E}_{P_n} \Delta_{\gamma_n}(f_\theta; \mathbf{H}) = \frac{1}{N} \sum_i \sup_{\mathbf{h}' \in \mathcal{X}} \left\{ \frac{\delta_{f_\theta}(\mathbf{h}')}{q(\mathbf{h}')} - \max_j \sup_{\mathbf{h}'' \in \mathcal{X}} \left\{ \frac{\frac{\delta_{f_\theta}(\mathbf{h}'')}{q(\mathbf{h}'')} - \frac{\delta_{f_\theta}(\mathbf{h}_j)}{q(\mathbf{h}_j)}}{c(\mathbf{h}_j, \mathbf{h}'')} \right\} c(\mathbf{h}_i, \mathbf{h}') \right\}.$$

It is easy to verify that

$$\mathbb{E}_{P_n} \Delta_{\gamma_n}(f_\theta; \mathbf{H}) \leq \frac{1}{N} \sum_i \sup_{\mathbf{h}' \in \mathcal{X}} \left\{ \frac{\delta_{f_\theta}(\mathbf{h}')}{q(\mathbf{h}')} \right\} + \frac{\delta_{f_\theta}(\mathbf{h}_i)}{q(\mathbf{h}_i)} - \sup_{\mathbf{h}'' \in \mathcal{X}} \left\{ \frac{\delta_{f_\theta}(\mathbf{h}'')}{q(\mathbf{h}'')} \right\} = \frac{1}{N} \sum_i \frac{\delta_{f_\theta}(\mathbf{h}_i)}{q(\mathbf{h}_i)},$$

as well as

$$\mathbb{E}_{P_n} \Delta_{\gamma_n}(f_\theta; \mathbf{H}) \geq \frac{1}{N} \sum_i \sup_{\mathbf{h}' \in \mathcal{X}} \left\{ \frac{\delta_{f_\theta}(\mathbf{h}')}{q(\mathbf{h}')} \right\} - \max_j \sup_{\mathbf{h}'' \in \mathcal{X}} \left\{ \frac{\frac{\delta_{f_\theta}(\mathbf{h}'')}{q(\mathbf{h}'')} - \frac{\delta_{f_\theta}(\mathbf{h}_j)}{q(\mathbf{h}_j)}}{c(\mathbf{h}_j, \mathbf{h}'')} c(\mathbf{h}_i, \mathbf{h}_j) \right\},$$

which also equals to $\frac{1}{N} \sum_i \frac{\delta_{f_\theta}(\mathbf{h}_i)}{q(\mathbf{h}_i)}$. Therefore, when $\gamma = \gamma_n$, we have $\mathbb{E}_{P_n} \left[ \Delta_{\gamma_n}(f_\theta; \mathbf{H}) \right] = \mathbb{E}_{P_n} \left[ \frac{\delta_{f_\theta}(\mathbf{H}_i)}{q(\mathbf{H}_i)} \right]$. Similarly, it can be shown that when $\gamma > \gamma_n$, the above equality also holds. Hence, we replace $\mathbb{E}_{P_n} \left[ \Delta_{\gamma_n}(f_\theta; \mathbf{H}) \right]$ with $\mathbb{E}_{P_n} \left[ \frac{\delta_{f_\theta}(\mathbf{H}_i)}{q(\mathbf{H}_i)} \right]$ in Theorem 1 and obtain the first result.

To obtain the second result, we define the transportation map [8]:

$$T_\gamma(f_\theta; \mathbf{h}) = \arg\max_{\mathbf{h}' \in \mathcal{X}} \left\{ \frac{\delta_{f_\theta}(\mathbf{h}')}{q(\mathbf{h}')} - \gamma c(\mathbf{h}, \mathbf{h}') \right\}.$$

Then according to (A.8), the empirical maximizer for $\sup_{\hat{P}: W_c(\hat{P}, P^*) \leq \rho} \int \delta(y, f_\theta(\mathbf{x}, \mathbf{z})) d\hat{P}(\mathbf{h})$ is attained by $\hat{P}(f_\theta) = \frac{1}{N} \sum_{i=1}^N I_{T_\gamma(f_\theta; \mathbf{h}_i)}$ where $I_{\mathbf{h}}$ assign point mass at $\mathbf{h}$, since it maximizes $\int \sup_{\mathbf{h} \in \mathcal{X}} \left( \frac{\delta_{f_\theta}(\mathbf{h})}{\hat{q}(\mathbf{h})} - \gamma c(\mathbf{h}, \mathbf{h}') \right) dQ_0(\mathbf{h}')$. Then we let $\rho_n(f_\theta) = W_c(\hat{P}(f_\theta), P_n)$, which equals to $\mathbb{E}_{P_n} \left[ c(T_\gamma(f_\theta; \mathbf{H}), \mathbf{H}) \right]$ by definition. So now we have

$$c_1 \gamma \rho_n(f_\theta) + \mathbb{E}_{P_n}[\Delta_\gamma(f_\theta; \mathbf{H})] = \sup_{P: W_c(P, P_n) \leq \tilde{\rho}} \mathbb{E}_P \left[ \delta(f_\theta; \mathbf{H}) / q(\mathbf{H}) \right],$$

for some $\tilde{\rho}$ that absorbs the excessive constant terms. We plug it into the Theorem 1 and obtain the second result. $\square$

## A.5 Implications from Tukey's Factorization on Unobserved Factors for Exposure

Here we discuss the Tukey's factorization which motivates our $G_\beta$ model that accounts for the unobserved factors in recommender system. Here we introduce the notation for the *counterfactual outcome* $Y_{u,i}(o)$, $o \in \{0, 1\}$, to denote what the user feedback would be if the exposure $O_{u,i}$ were given by $o$. In reality, we only get to observe $Y_{u,i}$ for either $O_{u,i} = 1$ or $O_{u,i} = 0$, and the tuple $(Y_{u,i}(1), Y_{u,i}(0))$ is never observed at the same time, which connects causal inference to the missing data literature. When there is no unobserved factor, the joint distribution of $(Y_{u,i}(1), Y_{u,i}(0))$ has simple formulation and can be estimated effectively from data using tools from causal inference. However, when unobserved factor exists, there are confounding between $(Y_{u,i}(1)$ and $Y_{u,i}(0))$, which

violates the assumption of many methods. The Tukey's factorization, on the other hand, characterizes our missing data distribution regardless of the unobserved factors as:

$$p_\beta\big(Y(o), O|\mathbf{X}, \mathbf{Z}\big) = p\big(Y(o)|O = o, \mathbf{X}, \mathbf{Z}\big)p\big(O = o|\mathbf{X}, \mathbf{Z}\big) \cdot \frac{p_\beta\big(O|Y(o), \mathbf{X}, \mathbf{Z}\big)}{p_\beta\big(O = o|Y(o), \mathbf{X}, \mathbf{Z}\big)}, o \in \{0, 1\},$$
(A.16)

where $\frac{p_\beta\big(O|Y(o), \mathbf{X}, \mathbf{Z}\big)}{p_\beta\big(O=o|Y(o), \mathbf{X}, \mathbf{Z}\big)}$ concludes the unknown mechanism in the missing data distribution [2, 1].

To see how the *counterfactual outcome* is reflected in the above formulation, when $O = \tilde{o} := 1 - o$ and $o = 1$, we have:

$$p_\beta\big(Y(1), O = 0|\mathbf{X}, \mathbf{Z}\big) = p\big(Y(1)|O = 1, \mathbf{X}, \mathbf{Z}\big)p\big(O = 1|\mathbf{X}, \mathbf{Z}\big) \cdot \frac{p_\beta\big(O = 0|Y(1), \mathbf{X}, \mathbf{Z}\big)}{p_\beta\big(O = 1|Y(1), \mathbf{X}, \mathbf{Z}\big)},$$

which gives the joint distribution of the outcome if the item was not exposed and the observed data where the item is exposed. Notice that both $p\big(Y(o)|O = o, \mathbf{X}, \mathbf{Z}\big)$ and $p\big(O = o|\mathbf{X}, \mathbf{Z}\big)$ can be estimated from the data, since $Y(o)$ is observed under $O = o$. So the only unknown mechanism in the missing data distribution is $p_\beta\big(O|Y(o), \mathbf{X}, \mathbf{Z}\big)/p_\beta\big(O = o|Y(o), \mathbf{X}, \mathbf{Z}\big)$.

Hence, we see the *counterfactual outcome* distribution can be given by:

$$p_\beta\big(Y(o)|O = 1 - o, \mathbf{X}, \mathbf{Z}\big) \propto p_{\text{obs}}\big(Y(o)|O = o, \mathbf{X}, \mathbf{Z}\big)/G_\beta\big(Y(o), \mathbf{X}, \mathbf{Z}\big), \quad o \in \{0, 1\}, \quad \text{(A.17)}$$

where $p_{\text{obs}}$ denotes the observable distribution and $G_\beta\big(Y(o), \mathbf{X}, \mathbf{Z}\big) = \frac{p_\beta\big(O=o|Y(o), \mathbf{X}, \mathbf{Z}\big)}{p_\beta\big(O|Y(o), \mathbf{X}, \mathbf{Z}\big)}$ characterizes the exposure mechanism even when unobserved factors exist. We treat the unknown $G_\beta\big(Y(o), \mathbf{X}, \mathbf{Z}\big)$ as a learnable objective in our setting. We have discussed in Section 3.2 that we use $g_\psi$ to characterize the role of $\mathbf{X}$ and $\mathbf{Z}$ in the exposure mechanism $G_\beta$, and hence we reach our formulation of $\delta\big(Y, f_\theta(\mathbf{X}, \mathbf{Z})\big)/G_\beta\big(Y, g_\psi(\mathbf{X}, \mathbf{Z})\big)$ in (8).

It has been discussed in [5] that including $Y$ in modelling the exposure mechanism may cause the self-selection problem in inference. Our setting does not fall into that category since our objective is to better learn the $f_\theta$ instead of making inferences. We also show in the following ablation studies that if the user feedback $Y$ is not included, i.e. $G_\beta\big(Y, g_\psi(\mathbf{X}, \mathbf{Z})\big) \equiv \sigma(g_\psi(\mathbf{X}, \mathbf{Z}))$, the improvements over the original models will be less significant.

## A.6  Experiment Settings and Complete Results

We provide the dataset details, preprocessing steps, train-validation-test split, simulation settings, detailed model configuration and implementation in this part of the appendix. We visualize the training process that reveals the adversarial nature of our proposed approach. We then provide the full ablation study and sensitivity analysis results to demonstrate the robustness of our approach.

### A.6.1  Real-world datasets

We consider three real-world datasets that covers movie, book and music recommendation.

- **Movielens-1M** [1]. The benchmark dataset records users' ratings for movies, which consists of around 1 millions ratings collected from 60,40 users on 3,952 movies. The rating is from 1 to 5, and a higher rating indicates more positive feedback.
- **LastFM** [2]. The LastFM dataset is a benchmark dataset for music recommendation. For each of the 1,892 listeners, they tag the artists they may find fond of over time. Since the tag is a binary indicator, the LastFM is an implicit feedback dataset. There is a total of 186,479 tagging events, where 12,523 artists have been tagged.
- **GoodReads** [3]. The benchmark book recommendation dataset is scraped from the users' public shelves on *Goodread.com*. We use the user review data on the *history* and *biography*

sections due to their richness. There are in total 238,450 users, 302,346 unique books, and 2,066,193 ratings in these sections. The rating range is is also from 1 to 5, a higher rating indicates more positive feedback.

### A.6.2 Data preprocessing and train-validation-test splitting

The Movielens-1M dataset has been filtered before made public, where each user in the dataset has rated at least 20 movies. For the LastFM and Goodread datasets, we first eliminate infrequent items (books/artists) and users that have less than 20 records. After examination, we find a small proportion of users having an abnormal amount of interactions. Therefore, we treat the users who have more than 1,000 interactions as spam users and not include them into our analysis.

The train-validation-test split is carried out based on the order of the user-item interactions. We adopt the standard setting, where for each user interaction sequence, all items but the last two are used in training, the second-to-last item is used in validation, and the last item is used in testing.

### A.6.3 Simulation settings

In a modern real-world recommender system, the exposure mechanism is determined by the underlying recommender model as well as various other factors. In an attempt to mimic the real-world recommender systems, we design a two-stage simulation method to generate the semi-synthetic data that remains truthful to real-world datasets. The first stage learns the characteristic from the data, such as the user relevance (rating) model and the partial exposure model (which may be inaccurate due to the partial-observation of exposure status), and the second stages simulates the working method of a real-world recommender system as well as user response. Since we wish to recover the user-item relevance as accurate as possible, we choose to use the explicit feedback dataset for our simulation, i.e. the Movielens-1M and Goodreads datasets.

In the first stage, given a true rating matrix, we train two hidden-factor matrix factorization models. The first model tries to recover the rating matrix and by minimizing the mean squared loss. We refer to this model as the *relevance model*. Since for the explicit feedback data the rated items must have all been exposed, so given the output $\hat{\mathbb{E}}[R_{u,i}|O_{u,i} = 1]$, we define the relevance probability as $p_{\text{sim1}}(Y_{u,i} = 1|O_{u,i} = 1) = \sigma\big(\hat{\mathbb{E}}[R_{u,i}|O_{u,i} = 1] - \epsilon_1\big)$, where $\sigma(\cdot)$ is the sigmoid function and $\epsilon_1$ reflects the perturbations brought by unobserved factors. The second model is an implicit-feedback model trained to predict the occurrence of the rating event, where instead of the original ratings, the non-zero entries in the rating matrix are all equal to 1. After obtaining the $\hat{p}(O_{u,i} = 1)$, we define the simulation exposure probability as $\log p_{\text{sim1}}(O_{u,i} = 1) = \log \hat{p}(O_{u,i} = 1) + \epsilon_2$, where $\epsilon_2$ also gives the extra randomness due to the unobserved factors. Now with the simulated $p_{\text{sim}}(Y_{u,i} = 1|O_{u,i} = 1)$ and $p_{\text{sim}}(O_{u,i} = 1)$ that reflects both the relevance and exposure underlies the real data generating mechanism as well as the effects of unobserved factors, we then generate the first-stage click data based by $p_{\text{sim1}}(Y_{u,i} = 1) = p_{\text{sim1}}(Y_{u,i} = 1|O_{u,i} = 1)p_{\text{sim1}}(O_{u,i} = 1)$.

In the first stage, we have generated an implicit feedback dataset that remains truthful to the original real dataset. Now we add the user-defined components that gives us more control over the exposure mechanism. Specifically, we obtain the new user and item hidden factors $\mathbf{x}, \mathbf{z}$ by training another implicit matrix factorization model using the generated click data. We generate the extra exposure function $e(\mathbf{x}, \mathbf{z})$, and add it to the $p_{\text{sim1}}$ and obtain the second-stage exposure mechanism $\log p_{\text{sim2}}(O_{u,i} = 1) = \log p_{\text{sim1}}(O_{u,i} = 1) + e(\mathbf{x}, \mathbf{z})$. The final click data is then generated by $p_{\text{sim2}}(Y_{u,i} = 1) = p_{\text{sim1}}(Y_{u,i} = 1|O_{u,i} = 1)p_{\text{sim2}}(O_{u,i} = 1)$.

Notice that having the second stage in the simulation is important, because the focus of the first stage is to mimic the generating mechanism of the real-world dataset. The second stage allows us to control the exposure mechanism. Also, we point out that retraining the implicit matrix factorization model in the beginning of the second stage is not required, thought it helps us to better characterize the data generated in the first stage.

### A.6.4 Model configuration and implementation

For all the baseline models we consider here (other than **Pop**), the dimension of the user and item hidden factors, initial learning rate and the $\ell_2$ regularization strength are the basic hyperparameters. We select the initial learning rate from $\{0.001, 0.005, 0.01, 0.05, 0.1\}$, and the $\ell_2$ regularization

strength from {0, 0.01, 0.05, 0.1, 0.2, 0.3}. The tuning parameters are selected separately to avoid excessive computations. We fix the hidden dimension at 32 for our models in order to achieve fair comparisons in the experiments. Also, notice that our approach has approximately twice the number of parameters with respect to the corresponding baseline model. In practice, the hidden dimension can be treated as a hyperparameter as well. We provide sensitivity analysis on the hidden dimension later in this section. We use the $Hit@10$ on validation data as the metric for selecting hyperparameters.

To make sure that the superior performance of our approach is not a consequence of higher model complexity, we double the hidden factor dimension of the baseline models to 64 when necessary.

Among the baseline models, the **Pop**, **CF** [6], **GMF** and **Neural CF** [3] are all standard approaches in recommender system who have relatively simpler structures, we adopt the default settings and do not discuss their details. We focus more on the attention-based sequential recommendation model **Attn** and the propensity-score method **PS**. For the **Attn**, we adopt the model setting from [9, 4] where the self-attention mechanism is added on top of a item embedding layer. We treat the hidden dimension of the key, query and value matrices, and the number of dot-product attention heads as the additional tuning parameters. For the **PS** method, there are two stages:

- Obtain $g_\psi^*$ by minimizing $\mathbb{E}_{P_n}\left[\delta\big(Y, g_\psi(\mathbf{X}, \mathbf{Z})\big)\right]$;

- Implement $\underset{f_\theta \in \mathcal{F}, \beta}{\text{minimize}} \mathbb{E}_{P_n}\left[\frac{\delta(Y, f_\theta(\mathbf{X}, \mathbf{Z}))}{G_\beta\big(g_\psi^*(\mathbf{X}, \mathbf{Z}), Y\big)}\right]$.

The tuning parameters for $g_\psi$ and $f_\theta$ are selected in each stage separately.

The configurations for the proposed approach consists of two parts: the usual model configuration for $f_\theta$ and $g_\psi$, and the two-timescale train schema. Firstly, we find out that the tuning parameters selected for $f_\theta$ and $g_\psi$ when being trained alone also gives the near-optimal performance in our adversarial counterfactual training setting. Therefore, we directly adopt the hyperparameters (other than the learning rate) selected in their individual training for $f_\theta$ and $g_\psi$. We experiment on several settings for the two-timescale update. Specifically, we wish to understand the impact of the relative magnitude of the initial learning rates $r_\theta$ and $r_\psi$. In practice, we care less about the learning rate discount when using the Adam optimizer, since the learning rate is automatically adjusted. Intuitively speaking, the smaller the $r_\psi$ (relative to $r_\theta$ ), the less $g_\psi$ is subject to the regularization in the beginning stage, and its adversarial behavior is less restricted. As a consequence, $f_\theta$ may not learn anything useful. We provide empirical evidence to support the-above point in Figure A.1, with the detailed discussion shown later. Finally, the regularization parameter $\alpha$ for the proposed approach is selected from {0.1, 1, 2}.

In conclusion, the hyperparameters that are specific to the proposed adversarial counterfactual training are the initial learning rates $r_\theta$ and $r_\psi$, as well as the regularization parameter $\alpha$.

### A.6.5   Computation

All the models, including the matrix factorization models, are implemented with *PyTorch* on a Nvidia V100 GPU machine. We use the *sparse Adam*[4] optimizer to update the hidden factors, and the usual Adam optimizer to update the remaining parameters. We use sparse Adam for the hidden factors because both the user and item factor are relatively sparse in recommendation datasets. The Adam algorithm leverages the momentum of the gradients from the previous training batch, which may not be accurate for the item and user factors in the current training batch. The sparse Adam optimizer is designed to solve the above issue for sparse tensors. We use the early-stopping training method both for the baseline models, where we terminate the training process when the validation metric stops improving for 10 consecutive epochs, and for our approach, where we monitor the minimax objective value and terminate the training process if it stops changing for more than $\epsilon = 0.001$ after 10 consecutive epochs.

It is straightforward to see that for a single update, the space and time complexity of our proposed adversarial counterfactual training is exactly the summation for that of $f_\theta$ and $g_\psi$ (where the complexity induced by $G_\beta$ is almost negligible). In general, our approach may take more training epochs to converge depending on the $r_{f_\theta}/r_\psi$ in our two-timescale training schema.

### A.6.6 Visualization of the adversarial training process

To demonstrate the underlying adversarial training process of the proposed adversarial counterfactual training method, we plot the training progress under several settings in Figure A.1 and A.2. From Figure A.1, we observe the following things.

- With a larger initial learning rate, $g_\psi$ tends to fit the data quicker than $f_\theta$.
- In the beginning, when $g_\psi$ has yet fit the data well, its adversarial behavior on $f_\theta$ is too strong, since both the loss value and the evaluation metric for $f_\theta$ is poor during that period. This also suggests the importance of using a larger initial learning rate for $g_\theta$.
- As the training progresses, $f_\theta$ eventually catches up with and outperforms $g_\psi$ in terms on the evaluation metric. However, the loss objective for $f_\theta$ is still larger, which is reasonable since it has the extra adversarial term in $\mathbb{E}_{P_n}\left[\delta\big(Y, f_\theta(\mathbf{X}, \mathbf{Z})\big)/G_\beta(Y, g_\psi(\mathbf{X}, \mathbf{Z}))\right]$ controlled by $g_\psi$. This also implies that $g_\psi$ is acting adversarially throughout the whole process.
- The training process gradually achieves the local minimax optimal, where either $f_\theta$ and $g_\psi$ are unable to undermine the performance of each other and their performance improves at the same pace in the later training phase.

Figure A.1: Adversarial training processes on the *Goodread* synthetic data using ACL (GMF / GMF) and ACL (MLP / MLP) as respectively. The upper panel gives the training objective for $f_\theta$ and $g_\psi$, i.e. $\mathbb{E}_{P_n}\left[\delta\big(Y, f_\theta(\mathbf{X}, \mathbf{Z})\big)/G_\beta(Y, g_\psi(\mathbf{X}, \mathbf{Z}))\right]$ and $\mathbb{E}_{P_n}\left[\delta\big(Y, g_\psi(\mathbf{X}, \mathbf{Z})\big)\right]$. The lower panel gives the evaluation metric on the validation dataset.

.

We then examine the adversarial training on the real-world dataset using the sequential recommendation model ACL (Attn / Attn). From Figure A.2, we first observe the same pattern as in Figure

A.1, which shows that the above discussions also apply to the real-world data and the sequential recommendation setting. Further, we conduct a set of experiment where the outcome is not included in modelling the exposure mechanism $G_\beta$. First of all, we see that the adversarial training patterns still hold whether or not we consider the outcome in $G_\beta$. Secondly, the performances, both in terms of the loss value and evaluation metric, are sub-optimal when $Y$ is not included in $G_\beta$.

Figure A.2: The adversarial training process on the real *Goodread* data using ACL (Attn / Attn) shows same pattern for the sequential recommendation setting, and demonstrates the effectiveness of including the outcome into the $G_\beta$ for modelling the exposure mechanism. The "use outcome" indicates whether $Y$ is used for modelling $G_\beta$.

### A.6.7   Complete ablation study

Due to the space limitation, we provide part of the ablation study in the main paper, and leave the rest to this part of the appendix. Firstly, we provide the complete results on using the propensity score model in Table A.2 for the three real-world datasets. By comparing with the results in Table 2, we see that our adversarial counterfactual training approach still outperforms their propensity score counterparts, which again emphasizes the importance of having the adversarial process between $f_\theta$ and $g_\psi$. Secondly, we provide the full set of results for the baseline models trained with our adversarial counterfactual approach on the real-world dataset (Figure A.2). As we mentioned in Section 5, models trained with our approach uniformly outperforms their counterparts. Notice that the superior performances of our approach do not benefit from a larger model complexity, since we have doubled the hidden factor dimension of the corresponding baseline models such that the number of parameters are approximately the same for all models.

### A.6.8   Sensitivity analysis

We provide the sensitivity analysis for the proposed adversarial counterfactual approach, mostly focus on the user/item hidden factor dimension size and the regularization parameter $\alpha$. We show the results of on the real-world datasets. The sensitivity analysis on user/item hidden factor dimension size is shown in Figure A.3, where we see that larger dimensions most often lead to better outcome (within the range we consider), which is in accordance with the common consensus in the recommender

| config | MLP Pop | MLP MLP | GMF Pop | GMF GMF | NCF Pop | NCF NCF | Attention Pop | Attention Attention |
|---|---|---|---|---|---|---|---|---|
| **MovieLens-1M** | | | | | | | | |
| Hit@10 | 61.93 (.2) | 60.85 (.1) | 64.21 (.3) | 62.19 (.1) | 63.78 (.4) | 61.28 (.2) | 81.97 (.1) | 81.05 (.2) |
| NDCG@10 | 33.37 (.1) | 31.90 (.2) | 34.96 (.1) | 32.53 (.2) | 34.05 (.1) | 30.98 (.3) | 54.51 (.1) | 52.33 (.1) |
| **Last-FM** | | | | | | | | |
| Hit@10 | 82.06 (.3) | 81.32 (.1) | 82.64 (.3) | 81.87 (.1) | 82.29 (.3) | 80.35 (.2) | 72.71 (.2) | 70.98 (.1) |
| NDCG@10 | 57.55 (.2) | 58.16 (.1) | 58.83 (.2) | 57.92 (.3) | 58.40 (.1) | 57.02 (.3) | 60.13 (.2) | 59.33 (.2) |
| **Goodreads** | | | | | | | | |
| Hit@10 | 62.59 (.1) | 60.03 (.3) | 64.92 (.2) | 64.43 (.2) | 63.75 (.2) | 61.44 (.3) | 73.39 (.3) | 71.37 (.2) |
| NDCG@10 | 38.01 (.2) | 37.32 (.1) | 39.21 (.1) | 38.45 (.1) | 38.85 (.2) | 38.03 (.1) | 49.99 (.1) | 49.18 (.2) |

Table A.2: Standard evaluations on the real-world data using the propensity-score models.

| ACL variant | ACL-MLP | | ACL-GMF | | ACL-NCF | | ACL-Attention | |
|---|---|---|---|---|---|---|---|---|
| **Metric** | Hit@10 | NDCG@10 | Hit@10 | NDCG@10 | Hit@10 | NDCG@10 | Hit@10 | NDCG@10 |
| *MovieLens-1M* | 62.04 (.2) | 33.59 (.2) | 64.32 (.2) | 33.70 (.1) | 63.97 (.2) | 34.81 (.1) | 83.64 (.1) | 55.71 (.2) |
| *Last-FM* | 82.88 (.2) | 57.43 (.2) | 83.64 (.2) | 59.11 (.1) | 83.09 (.2) | 58.93 (.2) | 72.02 (.2) | 59.45 (.1) |
| *Goodreads* | 62.90 (.2) | 38.57 (.1) | 64.57 (.2) | 39.54 (.1) | 63.95 (.2) | 38.72 (.1) | 73.82 (.3) | 49.99 (.1) |

Table A.2: Standard evaluations on the real-world data considering all ACL base model.

system domain. This also suggests that our approach inherits some of the properties from the $f_\theta$ and $g_\psi$, so the model understanding diagnostics also become easier if $f_\theta$ and $g_\psi$ are well-studied.

The sensitivity analysis on the regularization parameter $\alpha$ is provided in Figure A.4. We do not experiment on a wide range of $\alpha$; however, the results we have at hand already tells the patterns, that our approach achieves the best performances when $\alpha$ is neither too big nor too small. As a matter of fact, this phenomenon on regularization parameters is widely acknowledged in the machine learning community. In terms of our context, when $\alpha$ is too small, the regularization on $g_\psi$ becomes relatively weak compared with the loss objective of $f_\theta$, so $g_\psi$ does not fit the data well. As a consequence, $f_\theta$ also suffers from the under-fitting issues of $g_\psi$. On the other hand, when $\alpha$ gets too large, the minimax game will focus more on fitting $g_\psi$ to the data and overlooks $f_\theta$.

### A.6.9 Online experiment settings

The online experiments provide valuable evaluation results that reveal the appeal of our approach for real-world applications. All the online experiments were conducted for a content-based item page recommendation module under the implicit feedback setting where the users click or not click the recommendations. A list of ten items is shown to the customer on each item page, e.g. items that are similar or complementary to the anchor item on that page. The recommendation is personalized, so the user id and user features are included in the model as well.

In each iteration, new item features and user features are added into the previous model. The main architecture of the recommendation model is unchanged, which makes it favorable for examining our approach. By the time we write this paper, there have been four online experiments (AB tests) conducted for eight models that are trained offline using our proposed adversarial counterfactual training and then evaluated using the implicit feedback data. Unobserved factors such as the real-time user features, page layout and same-page advertisements are continually changing and not included in the analysis. The metric that we used to compare the different offline evaluation methods with online evaluation is the click-through rate.

Figure A.3: Sensitivity analysis of hidden factor dimension for the content-based ACL(GMF / GMF) model and the sequential ACL(Attn / Attn) model together with their corresponding baseline models, on the three real-world datasets. Recall that the hidden dimensions for the corresponding baselines are doubled from what is shown in the plots to achieve fair comparisons. From the top to bottom are results for the Movielens-1M, LastFM and Goodread.com data

Figure A.4: Sensitivity analysis on the regularization parameter $\alpha$ for the content-based ACL (GMF / GMF) model and the sequential ACL(Attn / Attn) model for their $f_\theta$ and $g_\psi$ components, on the three real-world datasets (from the top to bottom are results for the Movielens-1M, LastFM and Goodread.com data).

## Footnotes

[1]http://files.grouplens.org/datasets/movielens/ml-1m.zip

[2]http://files.grouplens.org/datasets/hetrec2011/hetrec2011-lastfm-2k.zip

[3]https://sites.google.com/eng.ucsd.edu/ucsdbookgraph/home

[4]https://agi.io/2019/02/28/optimization-using-adam-on-sparse-tensors/