[Reviews · NeurIPS 2020]

Review 1

Summary and Contributions: This paper argues to debias via an optimization framework that optimizes towards the worst case risk, which is a new idea in recommendation debiasing. The theoretical analysis also sounds interesting and is insightful.

Strengths: 1. Debiasing towards worst case exposure strategy is new 2. Theoretical analysis is interesting

Weaknesses: The novelty of the method seems to be limited, the author should compare to other similar works. 1. The worst case optimization framework is similar to DRO [1*]. The difference is that the author tries to optimize towards the worst case exposure strategy rather than group performance in DRO. 2. [2*] also proposes a dual learning algorithm to learn simultaneously the unbiased exposure distribution and the user preference. Some more experiments may be conducted against this type of work. In the experiments part, the paper's method shows minor improvement over POP as propensity weighting function. And I would like to see some explanations why in Table 1 g cannot be oracle in ACL, and why MLP/oracle is worse than MLP/Pop. [1*] Fairness Without Demographics in Repeated Loss Minimization, https://arxiv.org/abs/1806.08010 [2*] Unbiased Learning to Rank with Unbiased Propensity Estimation, sigir 2018

Correctness: Yes

Clarity: The notations are overwhelming, making it hard to concentrate on the backbone message that the authors try to deliver to this community.

Relation to Prior Work: Not really. Relationships with DRO and dual learning for debias are not discussed.

Reproducibility: Yes

Additional Feedback: For questions, see the weakness part. Overall, I found this paper interesting and insightful to read. I think it is debatable to say the exposure strategy is unknown when we take control of the exposure strategy in the system, i.e., though there are unobserved factors, these factors may not have impact on the exposure strategy that does not consider them.


Review 2

Summary and Contributions: In this paper, the authors study unbiased recommendation problem and propose an adversarial strategy to learn both recommendation and exposure models. Both theoretical and experimental analyses have been conducted to validate the effectiveness of the proposed method.

Strengths: 1. This paper studies an important problem. 2. The idea of adversary between the recommendation model and the exposure model is novel. How to learn IPS scores is an important problem but there are limited solutions. The proposed method is robust and has the potential to learn better IPS scores. 3. The authors provide rigorous theoretical analyses to validate the superiority of the proposed adversarial strategy.

Weaknesses: 1. Unclear Motivation. The authors give a statement that “the recommendation model is optimized over the worst-case exposure mechanism” but fail to give clear motivation behind the model. Why optimizing with the worst-case exposure is better than optimizing with the expected exposure that is widely adopted by existing methods? It seems that the essential advantage of the proposed method is robust. Uncertainty is not a good motivation as it has been considered by existing methods and can not answer the above question. 2. Insufficient experiments. The proposed method should be compared with existing unbiased recommendation methods (e.g. [a1][a2][32]) to validate the effectiveness of adversary. 3. Insufficient related works: 3.1 For the technical part, the formulation of the objective (eq.(4)) is a Wasserstein Robust Stochastic Optimization (DRSO) problem [a5][a6]. The difference in terms of solutions and generalization bounds between the paper with [a5][a6] should be discussed. 3.2 Some important related works on unbiased recommendation [a1][a2] or exposure-based recommendation are missing [a3][a4]. Also, the authors should better summarize the weakness of existing unbiased recommendation methods, as uncertainty has been considered by these methods. [a1] SIGIR'18: Unbiased Learning to Rank with Unbiased Propensity Estimation [a2] WSDM’20: Unbiased Recommender Learning from Missing-Not-At-Random Implicit Feedback [a3] CIKM’18: Modeling Users' Exposure with Social Knowledge Influence and Consumption Influence for Recommendation [a4] WWW’19: Samwalker: Social recommendation with informative sampling strategy [a5] Distributional Robustness and Regularization in Statistical Learning [a6] Regularization via Mass Transportation

Correctness: the claims and method are correct.

Clarity: Yes

Relation to Prior Work: Unclear. The details can refer to the weakness 3.

Reproducibility: Yes

Additional Feedback: I have read the author's response and increased my score. It's fine by me to accept the paper.


Review 3

Summary and Contributions: This paper investigates an important issue in recommender system regarding to the exposure bias. Despite that previous work address this problem through propensity-weighting approaches, this paper presents an interesting angle from adversarial learning to tackle the identifiability issue caused by implicit user feedback. The author derives learning bounds of the purposed minimax optimization problem and a robust offline evaluation metric through the introduced adversarial model. They evaluated their methods on simulated and real-world datasets and performed online experiments.

Strengths: * Interesting idea of leveraging adversarial training to tackle the exposure bias problem. * Theoretical grounds in the minimax optimization and counterfactual learning, derives an relaxation of the problem into a two-player game and corresponding learning bounds * This problem is relevant to the NeurIPS community and especially to the researchers of CI/RecSys 


Weaknesses: * Empirical evaluation of this paper is relatively weak and more discussions of the results are needed to support the claims the authors made. For example, most of the improvements are marginal on Goodreads and LastFM (the number scale in Table 1 is not consistent). Please include statistical tests for these results. Also, why ACL-MLP with a MLP exposure model outperform MLP with an Oracle exposure model, what does this indicate? * PS is a rather basic baseline for comparison. I would be curious to see how ExpoMF [1] and follow up works such as [2] compare to the purposed adversarial learning method * The organization of the paper could be improved. Some of the theoretical results could be referred to in the appendix and some results should be further discussed (e.g. Theorem 1) while more explanations on the empirical results could be discussed in the main paper. [1] Liang, D., Charlin, L., McInerney, J., & Blei, D. M. (2016 ). Modeling user exposure in recommendation. In Proceedings of the 25th international conference on World Wide Web [2] Wang, M., Gong, M., Zheng, X., & Zhang, K. (2018). Modeling dynamic missingness of implicit feedback for recommendation. In Advances in neural information processing systems

Correctness: yes

Clarity: yes

Relation to Prior Work: Please discuss how the local minimax ERM problem (referred to at line 144 and Equation 4) connects to/differs from the Wasserstein based Distributionally Robust Optimization [3,4]? [3] Duchi, J., & Namkoong, H. (2018). Learning models with uniform performance via distributionally robust optimization. arXiv preprint arXiv:1810.08750. [4] Ruidi Chen and Ioannis Ch Paschalidis. A robust learning approach for regression models based on distributionally robust optimization. The Journal of Machine Learning Research, 19(1):517–564, 2018.

Reproducibility: Yes

Additional Feedback:


Review 4

Summary and Contributions: This work claims that existing methods that rely on counterfactual modeling, make problem-specific or unjustifiable assumptions to bypass the identifiability issue. In contrast, this work utilizes the uncertainty brought by the identifiability issue and treat it as an adversarial component. Specifically, they propose a minimax objective function and optimize it over the worst-case exposure mechanism. By applying duality arguments and relaxations, they show that the minimax problem can be converted to an adversarial game between two recommendation models.

Strengths: 1. The approach is novel and the problem is interesting. Although there are some work use counterfactual learning approach from causal inference to address recommendation problem, they ignore the identifiability issues caused by partial-observation nature of user feedback data. Recent work on introducing adversarial modeling to solve the identifiability issue in observation studies focus mostly on learning balanced representation rather than the propensity-weighting method. 2. The authors propose a minimax objective function for counterfactual recommendation and convert it to a tractable two-model adversarial game. Furthermore, they prove the generalization bounds for the proposed adversarial learning and analyze the minimax optimization properties.

Weaknesses: 1. I think the key contribution of this work is the unjustifiable assumptions about identifiability for existing work. Unfortunately, the authors do not provide a theoretical analysis of the difference between the existing counterfactual learning on recommendation and the proposed method. The analysis of supervised learning on recommendation is not convincing. 2. The experimental analysis is nos sufficient; more concrete experiments are needed. It is very important to prove its effectiveness with SOTA methods (recommendation with casual inference), as well as show results in other standard datasets, which can show the effectiveness of the proposed with respect to different dataset complexities. More detailed analysis on the effect of the identifiability issue would be desirable. 3. No broader impact section.

Correctness: The paper seems to be correct.

Clarity: The paper is well written.

Relation to Prior Work: The related work is adequately discussed.

Reproducibility: Yes

Additional Feedback:

[Author Response · NeurIPS 2020]

We want to express our gratitude to all the reviewers for careful reading and valuable comments. We believe the advice will help us bring a better version of this paper. To begin with, we want to apologize for the typos and unclear writings. We will correct them in the final version, and add the broader impact section. It appears that there is a major concern regarding the contributions of our paper comparing with two lines of previous work, which we make clear first.

The first line of research is the various (unbiased) propensity estimation methods in the recommendation literature. The papers mentioned by the reviewers all assumes on a click model: $p(\text{click} = 1|x) = p(\text{expose} = 1|x) \cdot p(\text{relevance} = 1|x)$. Note that the implicit assumption being made is: $p(\text{expose} = 1, \text{relevance} = 1|x) = p(\text{expose} = 1|x) \cdot p(\text{relevance} = 1|x, \text{expose} = 1) = p(\text{expose} = 1|x) \cdot p(\text{relevance} = 1|x)$, so $p(\text{relevance} = 1|x, \text{expose} = 1) = p(\text{relevance} = 1|x)$, suggesting that relevance $\perp$ exposure $|\, x$, i.e. relevance is independent of exposure given the features. This may not be true (or at least cannot be examined) in many scenarios, unless we are able to collect every single factor that may affect the users' decision making process into $x$. The merit of our approach is that we get rid of the dependency on the click model assumption, and provide an alternative solution for researchers and practitioners who suspect the validity of relevance $\perp$ exposure $|\, x$ in their data. In practice, the gain from our approach depends on the degree of violation on the above assumption. Therefore, compare with the prior work, we introduce new perspectives and a feasible solution to this challenging problem. As for empirical evaluations, we have also tried our best to add more baselines according to the reviewers' requests (Table R.0). We ran into some trouble replicating the baselines with the published implementation, but we keep the our implementation as consistent as possible with the original work.

The other line of prior work is distribution-robust optimization (DRO), which is a vast domain. While our model also belongs to this category, the critical component that we argue for robustness is the propensity score distribution, which to the best our knowledge has not been studied before. The majority of papers in this domain have a different emphasis on the robustness of feature distribution or data generating distribution, which do not apply to our problem since the propensity score does not have a generative nature. From a technical perspective, the challenging part is that the propensity score term is placed on the denominator, so it requires extra proofs and arguments to obtain the duality, relaxation and concentration results. Our contribution is also novel in this regard.

**To Reviewer#1.** We thank Reviewer#1 for the insightful questions. We compare our work with other DRO and unbiased propensity estimation method as above and provide empirical comparisons with the mentioned baseline, where our approach still shows better performance. The reason why POP only give minor improvements in simulation is a consequence of the simulation setup where popularity is not directly related to exposure. The ORACLE methods may experience fluctuation because we have added random noise on the oracle to simulate the data, so ORACLE is only an unbiased estimation, but the variance can be large. Finally, we agree that the benefit of our approach is less significant when having access to the exposure strategy (which would be ideal), but this rarely happens in reality.

**To Reviewer#2.** We thank Reviewer#2 for pointing out the insufficiency of our manuscript, and we provide a refined analysis on the prior literature, including their weakness and comparisons with our work, in the above paragraph two. As we mentioned, uncertainty in exposure has not been well-handled by the unbiased propensity estimation methods, since they rely on another implicit assumption (the assumption for the click model) that may not be correct. Our approach provides an alternative solution that is free from the assumption. As for the empirical evaluations, we managed to add one set of additional experiment for the suggested baselines. It is possible that we have not tuned the baselines to perfection, but based on the initial result, the proposed approach still outperforms the propensity estimation approaches.

**To Reviewer#3.** We thank Reviewer#3 for the suggestions on further improvements. We apologize for the inconsistent scale in Table 1 where we forgot to multiply by 100 on Goodreads' results. Here, we provide a more detailed comparison with the mentioned literature in the beginning and add one set of experiment to include the SOTA baselines, where our approach still outperforms the propensity estimation approaches. As we mentioned in our response to Reviewer#1, the inconsistent performance of POP and ORACLE in simulation is a consequence of how we generate the data.

**To Reviewer#4.** We thank Reviewer#4 for the advice on providing a more in-depth comparison with the counterfactual recommendation literature and include more SOTA methods as baselines, which we provide at the beginning of this rebuttal. We wish to point out that the primary focus of recommendation is on the supervised learning part. Although we introduce a counterfactual learning component, the final model should still be examined in the supervised learning setting. The analysis for the identifiability issue, on the other hand, is a heated topic for the sensitivity analysis and would require another research paper to explore under the recommendation setting, which we pursue as future work.

| | MovieLens-1M simulation data | | | | | MovieLens-1M real data | | | | |
| --- | --- | --- | --- | --- | --- | --- | --- | --- | --- | --- |
| | URL-MF* | LtR* | ExpoMF | DM | ACL-MLP | URL-MF* | LtR* | ExpoMF | DM | ACL-GMF |
| Hit@10 | 16.33(.2) | 19.31(.4) | 16.26(.9) | 16.99(.8) | **21.58**(.1) | 63.71(.2) | 64.24(.1) | 62.50(.7) | 63.33(.8) | **64.32**(.2) |
| NDCG@10 | 7.26(.3) | 7.91(.3) | 7.24(.6) | 7.47(.5) | **8.42**(.2) | 33.19(.2) | 33.43(.1) | 32.85(.4) | 32.97(.5) | **33.70**(.1) |

Table R.0: Extra results on Movielens-1m simulation and real-world data. **URL-MF***: *Unbiased Recommender Learning from Missing-not-at-Random Implicit Feedback, WSDM'20* (the published code is not executable); **LtR***: *Unbiased Learning to Rank with Unbiased Propensity Estimation, SIGIR'18* (the published code is for search ranking); **ExpoMF**: *Modelling User Exposure in Recommendation, WWW'16*; **DM**: *Modelling Dynamic Missingness of Implicit feedback for Reommendation, NeurIPS'18*; **ACL-X**: the proposed adversarial counterfactual approach with model X as $f_\theta$ and $g_\psi$. Results have been multiplied by 100.

[Meta-Review · NeurIPS 2020]

Reviews were quite borderline, but ultimately slightly on the positive side. Given the borderline scores, a discussion was initiated. The reviewers raised some issues about novelty/comparisons (R1,R3), motivation (R2), missing work (R2,R3), and experimental analysis (R4). Mostly though the reviewers did not consider these to be critical issues. The discussion eventually resulted in some positive movement of the scores/comments and reached a consensus around recommending acceptance.